# The role of scaffold reshaping and disassembly in dynamin driven membrane fission

**Martina Pannuzzo[†], Zachary A McDargh[‡], Markus Deserno\***

Department of Physics, Carnegie Mellon University, Pittsburgh, United States

**Abstract** The large GTPase dynamin catalyzes membrane fission in eukaryotic cells, but despite three decades of experimental work, competing and partially conflicting models persist regarding some of its most basic actions. Here we investigate the mechanical and functional consequences of dynamin scaffold shape changes and disassembly with the help of a geometrically and elastically realistic simulation model of helical dynamin-membrane complexes. Beyond changes of radius and pitch, we emphasize the crucial role of a third functional motion: an effective rotation of the filament around its longitudinal axis, which reflects alternate tilting of dynamin's PH binding domains and creates a membrane torque. We also show that helix elongation impedes fission, hemifission is reached via a small transient pore, and coat disassembly assists fission. Our results have several testable structural consequences and help to reconcile mutual conflicting aspects between the two main present models of dynamin fission—the two-stage and the constrictase model.

DOI: https://doi.org/10.7554/eLife.39441.001

**\*For correspondence:**
deserno@andrew.cmu.edu

**Present address:** [†]Istituto Italiano di Tecnologia, Genova, Italy; [‡]Department of Chemical Engineering, Columbia University, New York, United States

**Competing interests:** The authors declare that no competing interests exist.

## Introduction

Of the three 80% homologous mammalian isoforms of dynamin, Dyn1, highly expressed in neurons, has been studied best (*Hinshaw, 2000*; *Praefcke and McMahon, 2004*). Crystallographic analysis (*Chen et al., 2004*; *Faelber et al., 2011*; *Ferguson et al., 1994*; *Ford et al., 2011*; *Zhang and Hinshaw, 2001*; *Chappie et al., 2010*) reveals five distinct domains: an N-terminal GTPase or G-domain; a 'stalk' region consisting of a four helix bundle; the signaling BSE domain that links G-domain and stalk; a pleckstrin homology (PH) domain binding phosphatidylinositol-4,5-biphosphate (PIP$_2$) lipids; and a proline-rich C-terminal domain (PRD) that mediates interactions with membrane scaffolding proteins. Dynamin monomers oligomerize via their stalks in a criss-cross fashion, resulting in stable dimers (*Cocucci et al., 2014*) or tetramers (*Ramachandran et al., 2007*) in solution. Continuing this assembly path yields helical filaments (*Carr and Hinshaw, 1997*), which have an outer diameter of about 50 nm and a helical pitch (i.e., height increment during one complete helical turn) between 10 nm and 20 nm in the absence of GTP (*Chen et al., 2004*). Specific interactions among dynamin subunits are not conserved throughout the large dynamin superfamily, yet all members share similar architectural assembly properties (*Praefcke and McMahon, 2004*).

Dynamin helices are narrower in the presence of GTP; the strongest constriction has been seen with the point mutant [K44A]Dyn1: it forms a stable superconstricted 2-start helix, tightening membrane tubules to a water-filled inner lumen with a diameter as narrow as 3.7 nm, but not cutting them (*Sundborger et al., 2014*). Since [K44A]Dyn1 is very inefficient in GTP hydrolysis, which however is necessary for remodeling (*Zhang and Hinshaw, 2001*; *Chappie et al., 2011*), this mutant is believed to trap the system in a pre-fission state. This leaves open the role of GTP activity, with presently two major competing scenarios: the two-stage and the constrictase model.

**eLife digest** When cells take up material from their surroundings, they must first transport this cargo across their outer membrane, a flexible sheet of tightly organized fat molecules that act as a barrier to the environment. Cells can achieve this by letting their membrane surround the object, pulling it inwards until it is contained in a pouch that bulges into the cell. This bag is then corded up so it splits off from the outer membrane. The 'cord' is a protein called dynamin, which is thought to form a tight spiral around the bag's neck, closing it over and pinching it away. The structure of dynamin is fairly well known, and yet several theories compete to explain how it may snap the bag off the outer membrane.

Here, Pannuzzo et al. have created a computer simulation that faithfully replicates the geometry and the elasticity of the membrane and of dynamin, and used it to test different ways the protein could work. The first test featured simple constriction, where the dynamin spiral contracts around the membrane to pinch it; this only separated the bag from the membrane after implausibly tight constriction. The second test added elongation, with the spiral lengthening as well as reducing its diameter, but this further reduced the ability for the protein to snap off the membrane. The final test combined constriction and rotation, whereby dynamin 'twirls' as it presses on the neck of the bag: this succeeded in efficiently severing the membrane once the dynamin spiral disassembled. Indeed, the simulations suggested that dynamin might start to dismantle while it constricts, without compromising its role. In fact, getting rid of excess length as the protein contracts helps to dissolve any remnants of a membrane connection.

Defects in dynamin are associated with conditions such as centronuclear myopathy and Charcot-Marie-Tooth peripheral neuropathy. Recent research also indicates that the protein is involved in a much wider range of neurological disorders that include Alzheimer's, Parkinson's, Huntington's, and amyotrophic lateral sclerosis. The models created by Pannuzzo et al. are useful tools to understand how dynamin and similar proteins work and sometimes fail.
DOI: https://doi.org/10.7554/eLife.39441.002

In the two-stage model (*Bashkirov et al., 2008*; *Mattila et al., 2015*), nucleotide-loaded dynamin assembles into highly constricted helices, whose entrapped lipid tubules spontaneously undergo hemifission; GTP hydrolysis induces subsequent detachment of the dynamin scaffold from the membrane, allowing the hemifission state to proceed to complete fission. However, neither the lack of dynamin's cooperativity with GTP during hydrolysis (*Tuma and Collins, 1994*), nor the short lifetime of the post-hydrolysis GDP + $P_i$ state (*Antonny et al., 2016*) are compatible with key requirements of the two-stage model. It also remains unclear why the experimentally observed superconstricted state does not reach hemifission. In the constrictase model (*Chappie et al., 2011*), several cycles of GTP hydrolysis induce adjacent turns (or 'rungs') to stepwise slide past one another, actively constricting the helix and its enclosed membrane tube until the latter fissions; disassembly is then a consequence of the vanishing lipid substrate. The problems here are that the cryo-EM density of stalks in a constricted (albeit: 2-start) helix does not match the X-ray structure of dimers, that it is not known how G-G links across rungs would unbind and step cooperatively, and that the mechanics of the final disassembly is less clear (*Antonny et al., 2016*).

To refine these two partially conflicting scenarios into a consistent and realistic model requires information about fast changes at the nanometer-scale. This is a challenge for all structural techniques capable of reaching the necessary resolution, and it often requires the use of mutants that might introduce artifacts. However, there is also a second problem: it is by no means obvious how a highly curved bilayer responds to the forces and torques imposed by such complex geometric constraints. Theoretical calculations have provided constraints on energetics and morphology (*Kozlov, 1999*; *Kozlovsky and Kozlov, 2003*; *Bashkirov et al., 2008*; *McDargh et al., 2016*; *McDargh and Deserno, 2018*), but these assume high symmetries and ignore fluctuations. Recent coarse-grained (CG) simulations, in which the dynamin helix is represented as a pair of rings, have elucidated the relevance of local torques (*Fuhrmans and Müller, 2015*) and the possibility of long-lived hemifission intermediates (*Mattila et al., 2015*; *Zhang and Müller, 2017*), but the implied

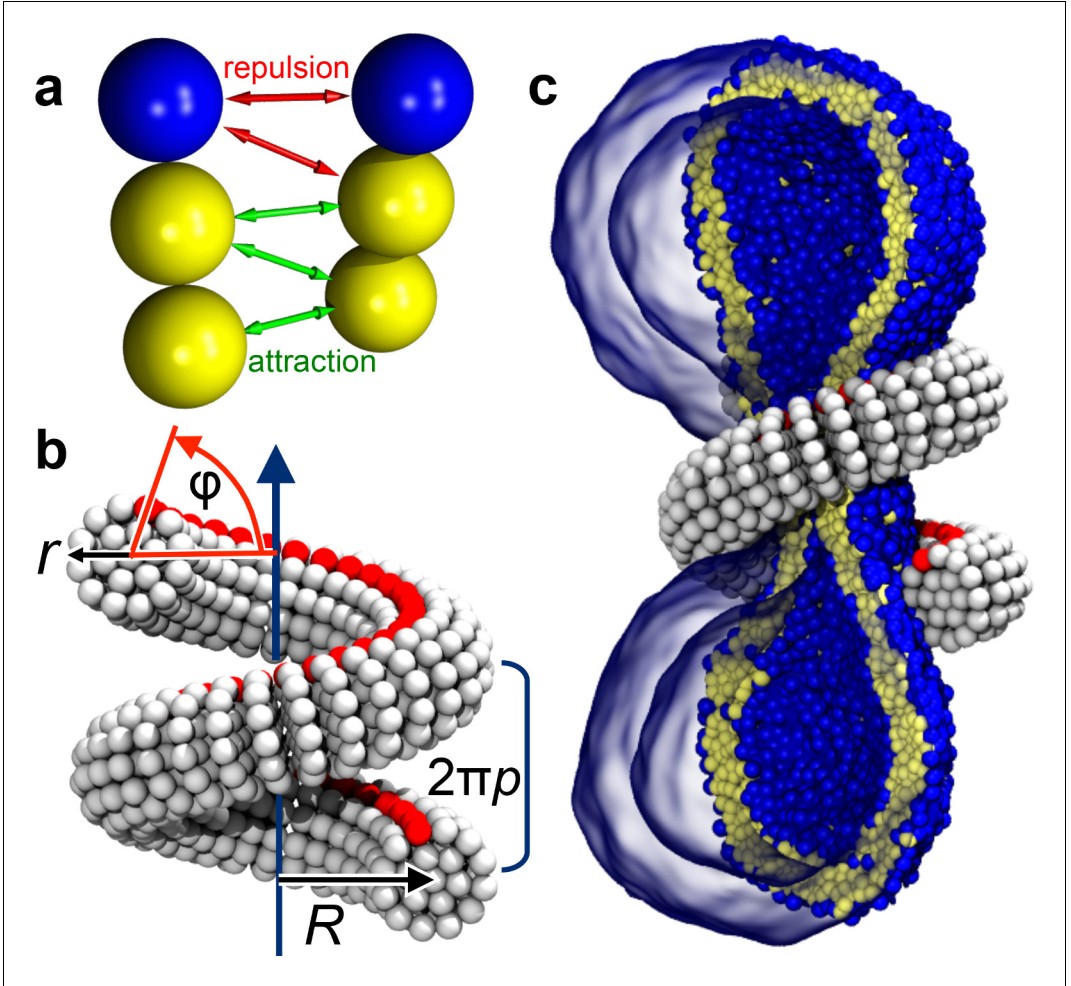

**Figure 1.** Ingredients of the model. (**a**) CG lipids comprising three beads assemble into membranes via tail bead attractions and head bead repulsions. (**b**) the dynamin helix is built as a stack of disks with 19 CG beads each; it contains an adhesive membrane-binding strip (red) that represents the PH domains and that can rotate by an angle $\varphi$ away from the inward-pointing direction. (**c**) a CG filament constricts the neck of a CG vesicle consisting of $\sim 10,000$ CG lipids.

DOI: https://doi.org/10.7554/eLife.39441.003

mirror symmetry differs from the actual helicoidal one. Here we go beyond these studies and investigate the constriction of fluctuating lipid bilayer tubes by geometrically realistic dynamin helices.

We employ an implicit-solvent CG membrane model (*Cooke et al., 2005*; *Cooke and Deserno, 2005*) (see Materials and methods) in which lipids are represented by three consecutive beads, each with a diameter of $\sigma \sim 0.8 \, \text{nm}$ (*Figure 1a*), that assemble via tail attraction and form bilayers with a bending rigidity of $\kappa \approx 13 \, k_{\text{B}} T$ (*Cooke et al., 2005*; *Cooke and Deserno, 2005*), where the thermal energy $k_{\text{B}} T = 4.1 \times 10^{-21} \, \text{J} \approx 0.6 \, \text{kcal/mol}$ provides the natural energy scale. The CG time scale maps to approximately $\tau \simeq 15 \, \text{ns}$, based on lipid self-diffusion, but we stress that this needs to be interpreted cautiously, since coarse-graining might speed up different dynamic processes differently. The emergent CG membranes capture not only a wide range of mesoscopic phenomena (*Deserno, 2009*), but also very local bilayer properties that likely play a role during leaflet-breaking fission events, such as a well-placed pivotal plane (*Wang and Deserno, 2015*), a correct magnitude and correlation length for lipid tilt (*Wang and Deserno, 2016*), and a pore-opening scenario (*Cooke et al., 2005*; *Deserno, 2009*) in agreement with previous simulations and continuum theory (*Farago, 2003*; *Tolpekina et al., 2004*). The dynamin filament is composed of similar CG beads, arranged and elastically connected to capture filament radius $r$, helical radius $R$ and pitch $2\pi p$, and

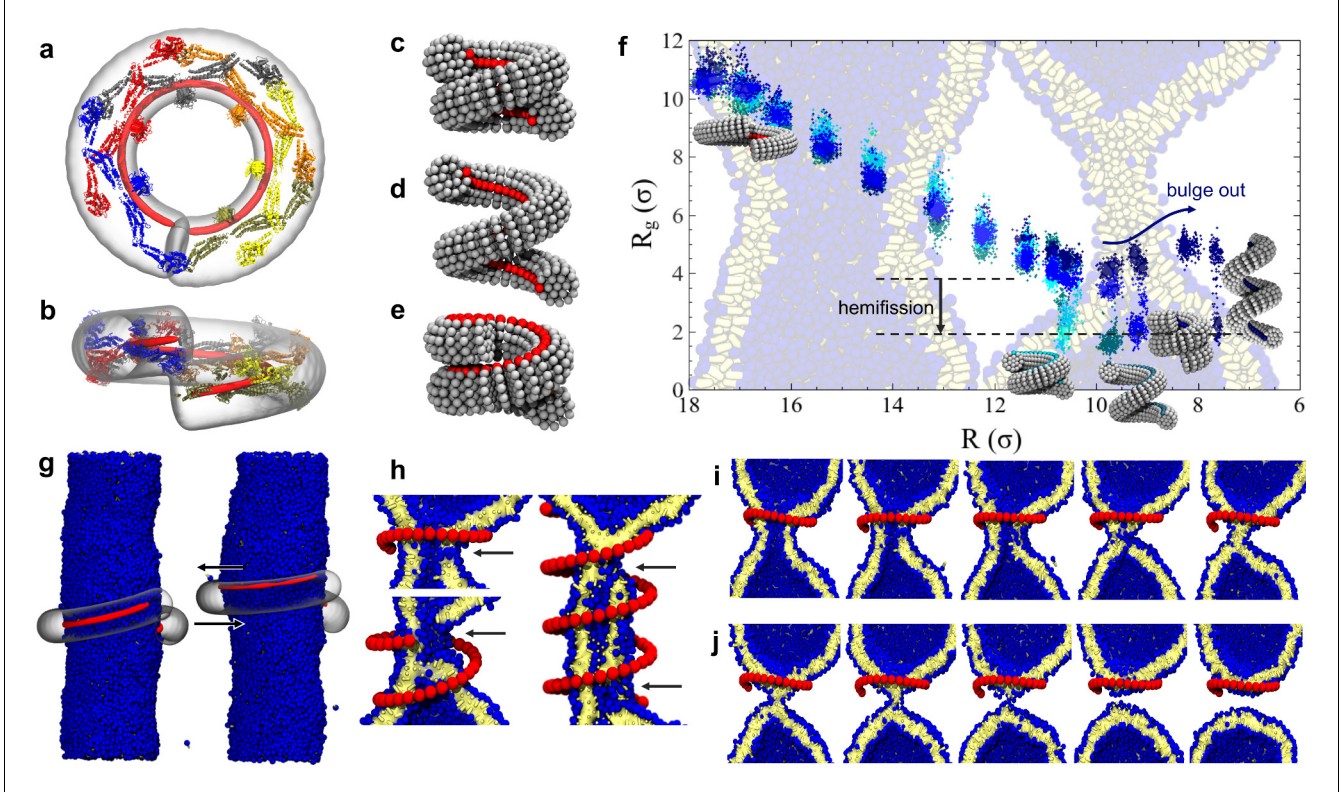

**Figure 2.** Dynamin shape changes and membrane response. (**a**) Top and (**b**) Side-view of dynamin dimers assembled into an unconstricted helical filament. (**c**) CG representation of a constricted filament. (**d**) Constricted and elongated filament. (**e**) Constricted and rotated filament. (**f**) Radius of gyration $R_\mathrm{g}$ of the membrane neck (see Materials and methods section) as a function of filament radius $R$ under four protocols: constriction only (blue) constriction+rotation (cyan), constriction+rotation+elongation (dark cyan), constriction+elongation (dark blue). (**g**) An unconstricted filament resting on a membrane of matching radius (left) creates a Darboux-torque once the (red) adhesion strip is rotated (right), inducing the membrane to asymmetrically bulge; the two arrows indicate the torque couple. (**h**) Cross-sectional view of a 1- and 1.5-turn helical scaffold at $R = 10.5\,\sigma$ and a 3.5-turn scaffold at $R = 10\,\sigma$. The filament was constricted and rotated, only the adhesion strip is shown. Hemifission seeds are small pores, visible as breaks in bilayer continuity (arrows). (**i**) Cross-cut illustration of membrane shape changes triggered by a simultaneous filament constriction, rotation, and gradual disassembly (***Video 5***), leading to hemifission. (**j**) Continuation of the previous sequence from hemifission to complete fission.
DOI: https://doi.org/10.7554/eLife.39441.004

The following source data and figure supplements are available for figure 2:

**Source data 1.** Constriction-elongation trajectory.
DOI: https://doi.org/10.7554/eLife.39441.010
**Source data 2.** Constriction-rotation-elongation trajectory.
DOI: https://doi.org/10.7554/eLife.39441.011
**Source data 3.** Constriction-rotation trajectory.
DOI: https://doi.org/10.7554/eLife.39441.012
**Source data 4.** Pure constriction trajectory.
DOI: https://doi.org/10.7554/eLife.39441.013
**Figure supplement 1.** Results of repeated simulation trajectories for the four different constriction protocols presented in ***Figure 2f***.
DOI: https://doi.org/10.7554/eLife.39441.005
**Figure supplement 1—source data 1.** Constriction-elongation trajectories.
DOI: https://doi.org/10.7554/eLife.39441.006
**Figure supplement 1—source data 2.** Constriction-rotation-elongation trajectories.
DOI: https://doi.org/10.7554/eLife.39441.007
**Figure supplement 1—source data 3.** Constriction-rotation trajectories.
DOI: https://doi.org/10.7554/eLife.39441.008
**Figure supplement 1—source data 4.** Pure constriction trajectories.
DOI: https://doi.org/10.7554/eLife.39441.009

the location of an effective PH binding strip (*Figure 1b* and Figure 5, Materials and methods). In the present study we forgo the dynamin assembly process and start with a pre-formed helix winding around the neck of an elongated vesicle comprising about 10,000 lipids (*Figure 1c*). We change the scaffold geometry by tuning the equilibrium distances between its CG beads, that is by imposing internal stresses that trigger a global elastic shape relaxation, and we do this slowly enough to avoid viscous stresses in the bilayer (Figure 6, Materials and methods). All simulations were performed using the ESPResSo package (*Limbach et al., 2006*) and run in triplicate, leading to consistent results. Visualization was done with VMD (*Humphrey et al., 1996*).

## Results

### Pure constriction

We begin by exploring a process in which the filament constricts its helical radius $R$ but maintains both the pitch and the orientation of the PH domain with respect to the substrate (*Figure 2c*). At a confinement approximately corresponding to the superconstricted state, the enclosed membrane tube remains stable. The hemifission intermediate only forms at constriction levels that eliminate the inner lumen (*Figure 2f*, blue curve, and *Video 1*). This seems to contradict the findings by *Kozlovsky and Kozlov (2003)* that an inner lumen diameter smaller than approximately 2 nm should spontaneously proceed towards hemifission, but this is not so: in their study an approximately catenoidal neck fissions by the application of bending moments at the upper and lower edge of the neck, while in our case the neck is created by external constriction—a mechanically different scenario. It is of course conceivable that our bilayer tube is merely kinetically stable, but nothing analytical is known about the barrier towards the topologically distinct hemifission state. Indeed, the dynamin-coated superconstricted lipid tubule is stable in experiment (*Sundborger et al., 2014*).

### Constriction and elongation

Some experiments have shown that GTP hydrolysis may increase the filament's pitch at fixed radius (*Figure 2d*), which has led to the suggestion that dynamin might act as an extension

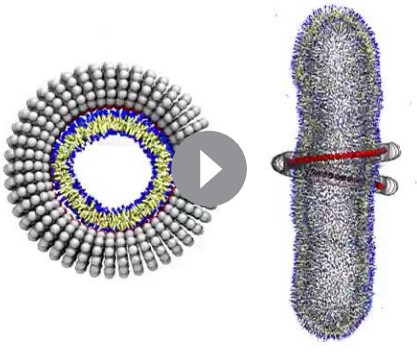

**Video 1.** Pure filament constriction. A dynamin filament, initially with radius $R = 18\,\sigma$, pitch $2\pi p = 12\,\sigma$ and rotation angle $\varphi = 0$ is constricted by stepwise reduction of the radius $R$ down to the critical radius $R = R_c = 10.5\,\sigma$, following the blue time-series in *Figure 6*. The right part of the video shows a side view, in which the 10,000 lipids of the vesicle are only rendered in stick-representation such as to permit some amount of transparency. The dynamin filament is shown as a transparent gray tube, and only the adhesive PH domain strip is explicitly shows as a sequence of red beads. The left part of the video shows a view along the helical axis of the filament, and lipids are only shown if their $z$-distance from the filaments center of mass is within $\pm 1\,\sigma$, such as to follow the extent of constriction and the diameter of the luminal region.

DOI: https://doi.org/10.7554/eLife.39441.014

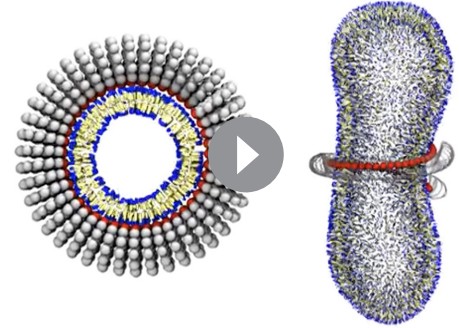

**Video 2.** Filament constriction and rotation. Same as *Video 1*, but this time the filament is additionally rotated, following a combination of the blue and red time series in *Figure 6*. The hemifission event towards the end happens at $R = R_c$, at which the non-rotation protocol shown in *Video 1* does not lead to hemifission. It results in a stable cylindrical micelle spanning the two daughter vesicles.

DOI: https://doi.org/10.7554/eLife.39441.015

spring ('poppase model') (*Stowell et al., 1999*). However, since dynamin changes its shape slowly compared to a membrane's viscoelastic time scale, believed to be shorter than about 10 ms (*Camley and Brown, 2011*), this putative mechanism cannot rely on viscous stresses but instead only on the reversible work being performed during elongation of the neck. To test this, we simultaneously decrease the radius and increase the pitch (Materials and methods, Figure 6). Contrary to expectation, helical elongation *impedes* fission: at the same helical scaffold radius $R$, a larger pitch results in a larger effective radius $R_g$ of the enclosed membrane tubule, requiring further constriction to achieve hemifission. This happens because a large pitch prevents the helical filament from symmetrically confining the membrane, allowing it to 'bulge out' at the open groove (*Figure 2f*, dark blue curve). In nature this could be prevented by accessory proteins, or—if the two-start helix is physiologically relevant—by the second interlocking filament. Compared to the pure constriction case, which in all three runs transitioned into the hemifission step at the same constriction step, we here observe a slight scatter of the precise transition points between two consecutive steps—see *Figure 2—figure supplement 1d*. We attribute this to the reduced confinement of a membrane inside a helical scaffold with a widening groove, permitting more fluctuations that render the transition point less definite.

## Constriction and rotation

Beyond changes of radius and pitch, the set of macroscopic motions available to a helical filament contains a third possibility: a 'twirling' motion around the local filament axis (*Figure 1b* and *Figure 2e*). Its relevance for dynamin derives from a growing number of studies that provide experimental evidence for the tilting of dynamin's PH domains. *Mehrotra et al. (2014)* have shown that a change in PH domain orientation may regulate fission. *Sundborger et al. (2014)* fitted dynamin's crystallographic structure to EM density maps and noticed that one of the two PH domains per dimer tilts out of the membrane, thereby breaking the symmetry of the dimer. And a very recent 3.75 Å resolution Cryo-EM reconstruction of human Dyn1 in the GMPPCP-bound state (*Kong et al., 2018*) shows that the bundle signaling element (BSE) is asymmetrically bent, presumably due to forces generated from the GTPase dimer interaction. These forces are further transferred across the stalk to the PH domain and onto the lipid membrane.

On mesoscopic scales, this local rearrangement of the tertiary structure can *effectively* be described as a net *rotation* of the helical scaffold around its local longitudinal axis, even if the core of the protein filament does not actually co-rotate. If the PH domains of each monomer were to tilt in the same fashion, then due to dynamin's criss-cross assembly half of the PH domains would tilt 'up' while the other half would tilt 'down', which on average does not displace the filament's binding surface. However, if only one of these two sets of PH domains tilts, this breaks the up-down symmetry and—at the level of our CG model—effectively rotates the average position of the adhesive strip away from the substrate. This creates a tangential torque on the membrane, because adhesion pins the filament's local material frame to its Darboux frame on the surface (*Guven et al., 2014*), and therefore it has appropriately been called 'Darboux torque' (*Fierling et al., 2016*)—see *Figure 2g* and *Box 1*.

Considering that the PH domain is connected to the stalk rather flexibly, it is not obvious that it can actually transmit such a torque. However, flexibility does not preclude the transmission of stresses, provided other contacts are in place that help acting as a pivot, and these might simply be steric interactions with other parts of the assembled scaffold. We recall that *Kong et al. (2018)* provide structural evidence that forces are transmitted all the way from dynamin's G-domain to the PH domain and onto the membrane, even though this is indirectly deduced by aligning the coordinates of dynamin with an unbent BSE to the structure of a bent dynamin in the Dyn$^{GMPPCP}$ map at the GTPase domain. Still, neither this new structure, nor the previously observed tilting of the PH domain (*Sundborger et al., 2014*) unambiguously proves the existence of torques. But while elucidating the mechanical nature of this process will have to await more detailed molecular modeling, the existence of a Darboux torque as been posited and exploited in another model for dynamin-driven fission, which simplifies the complex scaffold geometry to two counter-rotating apposing rings (*Fuhrmans and Müller, 2015*; *Shnyrova et al., 2013*), a geometry that remote-pinches the membrane between the rings. Notice, though, that both geometry and symmetry are different in the latter case: counter-rotating rings have a mirror symmetry and hence exert *oppositely* acting Darboux torques that constructively interfere in the middle of the scaffold. Since a helix is one

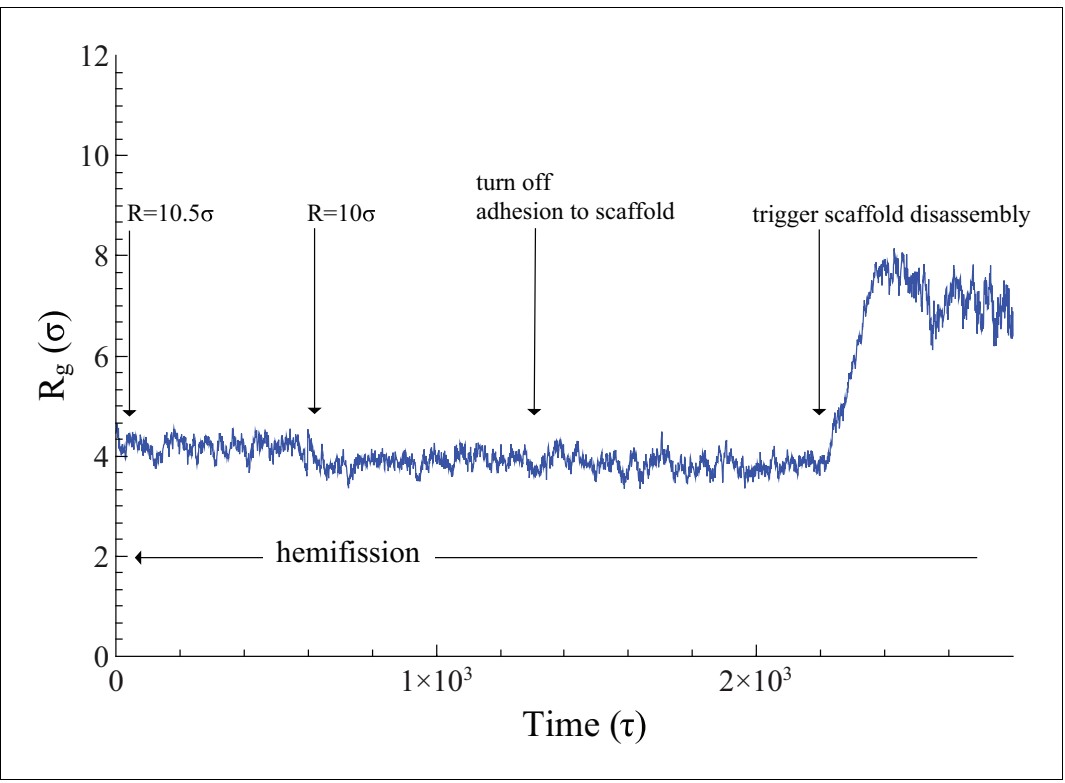

**Figure 3.** Time evolution of $R_g$. The neck's gyration radius is shown in the end stages of a pure constriction protocol. The time series starts when the scaffold radius is $R = R_c = 10.5\,\sigma$, which leads to $R_g/\sigma = 4.187 \pm 0.02$ (as measured over the subsequent $600\,\tau$, with an error determined via blocking (**Flyvbjerg and Petersen, 1989**)). After that, a further constriction of the filament to $R = 10\,\sigma$ reduces the neck's gyration radius to $R_g/\sigma = 3.890 \pm 0.04$ but does not trigger hemifission (which corresponds to the much smaller value $R_g \approx 2\,\sigma$—see **Figure 2f**). Turning off the adhesion between scaffold and membrane only reduces the gyration radius by a very minor amount, $R_g/\sigma = 3.841 \pm 0.035$, a change that is not statistically significant ($p = 0.36$). Once we additionally let the scaffold disassemble into (non-adhesive) dimers (see also **Video 4**), the neck very rapidly doubles its radius within about $200\,\tau$, after which the definition of its location becomes ambiguous.

DOI: https://doi.org/10.7554/eLife.39441.018

The following source data is available for figure 3:

**Source data 1.** Gyration radius as a function of time.
DOI: https://doi.org/10.7554/eLife.39441.019

contiguous filament, it will rotate everywhere in the same sense, rendering torque amplification much less straightforward.

In our own simulations, which account for the full helical geometry, we find that rotation is necessary for a constricted membrane tube with a remaining aqueous lumen to transition into the hemifission intermediate. Consistent across all three simulation runs, the constriction plus rotation sequence (**Figure 2f**, cyan curve, **Figure 2—Figure Supplement 1a**, and **Video 2**) transitions at $R = R_c \approx 10.5\,\sigma \approx 8.4\,\text{nm}$, when the inner lumen is small but has not yet vanished; we will subsequently refer to $R_c$ as the 'critical constriction radius'.

Since rotation gradually diminishes binding between membrane and filament, a non-rotating scaffold maintains adhesion while constricting, which could hold the membrane tube open and thereby prevent it from transitioning into the hemifission case. To test this, we considered a constricted but *not* rotated filament at $R = 10\,\sigma$ (which is even below the critical radius) and artificially switched off the adhesion between scaffold and membrane. As shown in **Figure 3**, the neck did not progress towards hemifission; in fact, the response to this change is a reduction in $R_g$ which, considering the error bars (determined via blocking (**Flyvbjerg and Petersen, 1989**)) is so small that it is not statistically significant. This suggests that the functionally important consequence of rotation is not merely

a reduction of binding energy, but active stresses, likely due to the above-mentioned Darboux torque.

If in addition to constriction and rotation we also elongate the helix, hemifission is still reached before the inner lumen disappears, but this requires a further filament radius reduction (*Figure 2f*, dark cyan curve). Consistent with our observations on constriction plus elongation, this case with additional rotation also shows a greater variability across our three runs, transitioning over three consecutive constriction steps, which however fall between the bounds of constriction plus rotation and constriction only (see *Figure 2—figure supplement 1b*). These findings again endorse the view that rotation supports hemifission, while elongation opposes it.

## Pre-hemifission disassembly of the dynamin coat

In the two-stage model the constricted state assembles from GTP-loaded dynamin, while subsequent hydrolysis-triggered depolymerization of the filament induces fission (*Bashkirov et al., 2008*; *Mattila et al., 2015*). Two pathways are conceivable: pre-hemifission coat disassembly would need to drive the membrane tube towards (at least) hemifission, while post-hemifission disassembly would only need to destabilize this hemifission state. To test the first pathway, we start with a filament at $R = R_c$ (induced only via constriction) and then cut it into approximately dimer-sized fragments. Irrespective of whether these retain their adhesion capability to the membrane (*Video 3*) or lose it and hence immediately unbind (*Video 4*), the membrane neck rapidly widens following scaffold rupture (see also *Figure 3*). Of note, even binding-capable fragments are individually not strong enough to impose their curvature on a substrate whose geometry becomes progressively unfavorable, and they ultimately detach from it. This also documents that our scaffold only binds weakly, and that rotation-driven hemifission is not the trivial result of massive forces resulting from large adhesion, or even of pulling lipids out of the membrane.

Observe that our simulation setup does not exert external membrane tension, since our implicit solvent enforces no volume constraint and permits the membrane—even though closed and constricted—to relax the area per lipid. Nevertheless, it is unlikely that a nonzero tension would prevent the widening of the neck: the value that would be needed to maintain a cylindrical membrane at the critical radius $R_c$ is given by $\kappa/2R_c^2$, or a few mN/m (*Deserno, 2015*; *Bukman et al., 1996*; *Hochmuth et al., 1996*), which is very high. Even disregarding the question how such a large tension would arise in a physiological context, the mere fact that this approaches a membrane's rupture stress and would hence place the system dangerously close to unspecific bilayer failure renders a recourse to such a high tension implausible.

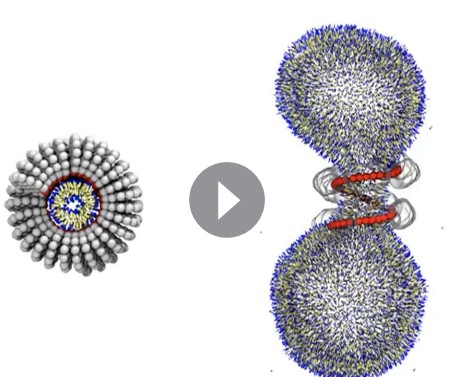

**Video 3.** Filament breakage with retained adhesion. A filament constricting a vesicle down to the critical radius $R_c$ is suddenly cut into pieces comprising five discs each (corresponding roughly to dynamin dimers). Even though the membrane curving fragments retain their adhesion energy with respect to the membrane substrate, this leads to a re-expansion of the neck radius, followed later by fragment unbinding once the substrate curvature becomes too unfavorable.
DOI: https://doi.org/10.7554/eLife.39441.020

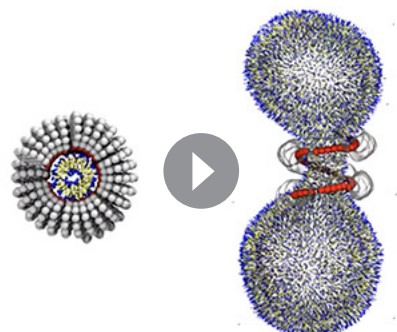

**Video 4.** Filament breakage with simultaneously lost adhesion. Same as *Video 3*, except that the fragments also lose their binding affinity to the membrane, causing them to immediately detach and stop curving the substrate.
DOI: https://doi.org/10.7554/eLife.39441.021

## Box 1. Illustration of a tangential membrane torque.

An object in contact with a membrane can exert a *force* on it by locally pushing or pulling. But it can also exert a *tangential torque* (*i.e.*, a torque whose rotation axis is parallel to the membrane plane). This can be conceptualized as a *force couple*—two forces of equal magnitude but opposite direction, which do not share the same line of action. As an illustration, panel (a) shows a pen clipped to a flat piece of paper and held tangentially to its surface. Rotating the pen around its axis, as in (b), exerts a horizontal torque on the paper pinned between pen and clip, resulting in an S-shaped curvature deformation. The rotation is illustrated as a blue curve, the corresponding force couple as two red arrows. Darboux torques are tangential torques that are not triggered by an explicit external rotation (as in the pictured example) but through a mismatch between the direction of a spontaneously curved filament's normal curvature and the direction of its adhesive strip (*Fierling et al., 2016*), which in our case is quantified by the angle $\varphi$ (see *Figure 1b*). Mechanical equilibrium then also requires that in the absence of any external forcing the *total* Darboux torque integrates to zero.

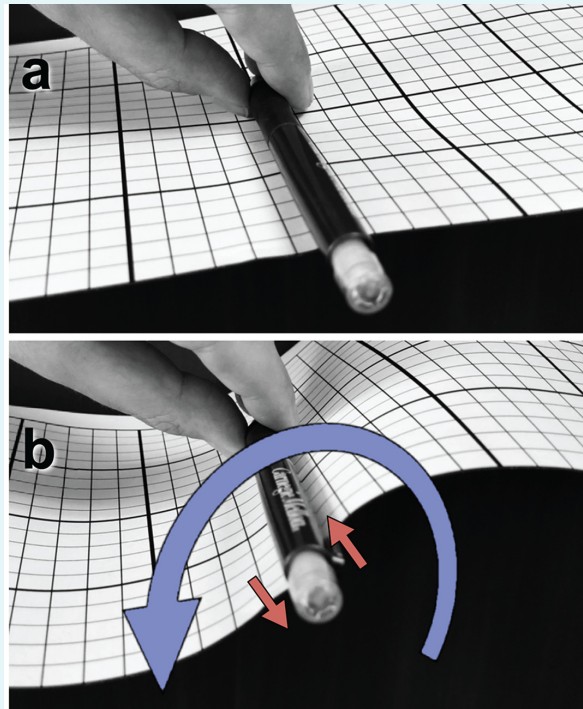

**Box 1—Figure 1.** Illustration of a tangential membrane torque.

DOI: https://doi.org/10.7554/eLife.39441.016

DOI: https://doi.org/10.7554/eLife.39441.017

## Non-axisymmetric pathway to initiate hemifission via pores

Recent experiments have observed that fission of dynamin-coated membrane tubules occurs within the coated region (*Dar et al., 2015*), although earlier studies had suggested that fission occurs at the edge between the dynamin helix and the uncoated membrane (*Morlot et al., 2012*). We always observe that an initial hemifission seed appears in the middle of the neck for short filaments (up to 1.5 turns), or two seeds appear near the edges for longer ones (more than two turns), see *Figure 2h*. This suggests that two regions of highly localized stress appear slightly inwards of either edge, which overlap and may mutually amplify for sufficiently short filaments. Furthermore, since

longer filaments trigger hemi-fission only at an even smaller scaffold radius ($R_{c,\,long} \approx 10\sigma$), both findings indicate that short filaments induce hemifission more efficiently than long ones.

That hemifission is seeded by a small pore is unexpected, not only because fission is believed to be non-leaky (*Bashkirov et al., 2008*), but also because the existence of a hemifission intermediate is generally taken to exclude the need for (or even possibility of) pores. However, a tube's inner leaflet must change topology, posing the question how a filament only in contact with the outer leaflet could promote this transition. Our simulations suggest an intriguing non-axisymmetric pathway: the filament seeds a small pore that widens around the circumference of the neck, while its top and bottom edges pull radially inward and fuse with the apposing intact membrane. This creates two defect lines at which three bilayers meet, and contracting them results in the top and bottom point singularity of the hemifission state's cylindrical micelle. This sequence of events never involves a large pore, *provided* the filament stays short enough, a condition that imposes constraints on the constriction process itself.

## Concomitant constriction and disassembly of the scaffold

If dynamin constricts due to GTP hydrolysis, as posited in the constrictase model, the initially unconstricted helix has to be long enough for the two rungs to meet, so that apposing G-domains can cross-catalyze hydrolysis. Constricting the radius by a factor of two then necessarily doubles the number of turns if the helical length stays fixed. We find that such long scaffolds induce rather sizable pores (*Figure 2h*) that would likely result in leakage. But the ends of a filament that in the constricted state only takes one turn would not be able to meet up in the preceding unconstricted state, and GTP driven constriction could not commence. This dilemma could be resolved if constriction and disassembly happen in short succession: when the two ends of the initially relaxed helical filament first meet, they hydrolyze GTP and trigger a first constriction step. This brings a new pair of GTP loaded G-domains in contact that drive the next constriction step, and so forth until full constriction. But since GTP hydrolysis enhances disassembly, a filament end comprising ever longer stretches of spent dynamin can shed its monomers concurrently with the ongoing constriction, especially if its PH domain retracted from the membrane. The helix would tighten but never

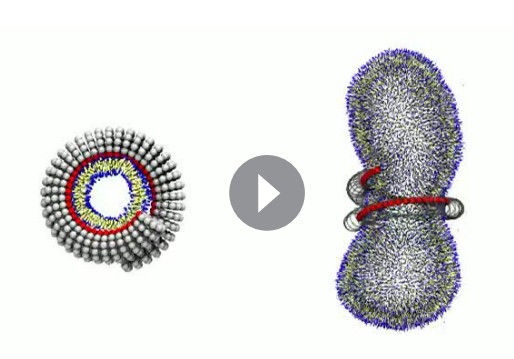

**Video 5.** Constriction, rotation, and concurrent filament depolymerization. Same as *Video 2*, except that the filament also concurrently depolymerizes during constriction and rotation. As a result, once hemifission sets in, the two point-defects at the ends of the resulting cylindrical hemifission micelle are no longer kept far apart by the now much shorter enclosing scaffold, permitting the defects to annihilate and thus fission to complete.
DOI: https://doi.org/10.7554/eLife.39441.022

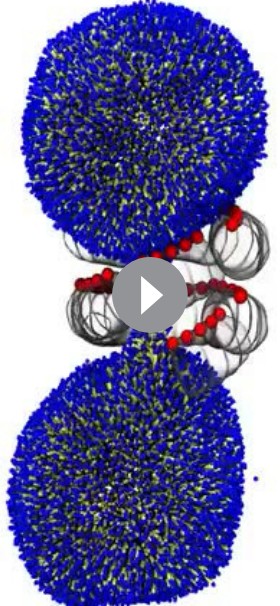

**Video 6.** Fission via unbinding and defect merger. If at the endpoint of *Video 2* the filament depolymerizes, the cylindrical hemifission micelle pulls the two daughter vesicles together and permits the point defects to merge and annihilate, thus completing fission.
DOI: https://doi.org/10.7554/eLife.39441.023

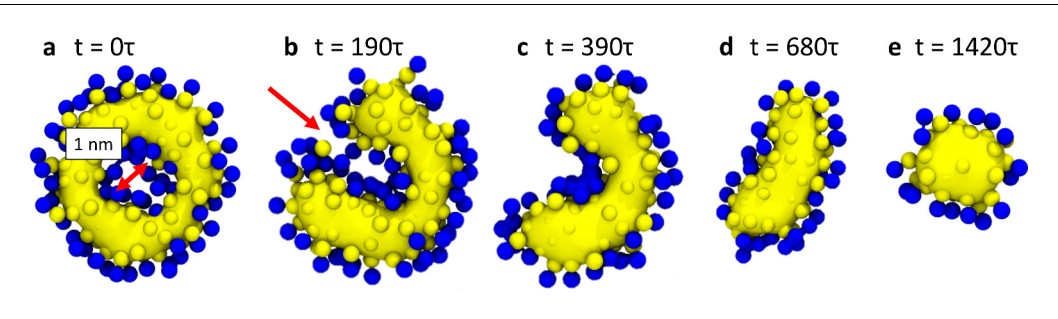

**Figure 4.** Hemifission via a transient pore. This time sequence of simulation snapshots shows slices through the membrane neck with a width of $2\sigma$, placed symmetric around the center of mass of the scaffold (not shown for clarity). The continuity of the tail region (yellow; blue are the head groups) is emphasized by using VMD's 'QuickSurf' rendering on all tail beads (*Humphrey et al., 1996*), which creates an isosurface extracted from a volumetric Gaussian density map. In panel (**a**), chosen to be time-point $0\tau$ in this sequence, the tail region is still continuous and encloses an inner lumen with a diameter of about $1\,\mathrm{nm}$. In panel (**b**) a small pore opens (red arrow) that connects the inner lumen to the exterior region of the vesicle. As it widens through panels (**c**) and (**d**), the pore rim above and below the imaged plane fuses to the inner leaflet of the lumen, which finally leaves in panel (**e**) a cylindrical hemifission micelle connecting two closed vesicles. See also *Video 2*.
DOI: https://doi.org/10.7554/eLife.39441.024

extend much beyond one turn, and hence never induce large pores (*Video 5*). This hypothesis is compatible with the experimental evidence that, upon GTP hydrolysis, the scaffold seems to adjust to a length optimal for fission (about one full turn) (*Shnyrova et al., 2013*).

## Transition from hemifission to complete fission

Without concurrent filament shortening, our hemifission state resembles a short cylindrical micelle (*Figure 2f*, background image), a transition state explored in two other recent investigations (*Mattila et al., 2015*; *Zhang and Müller, 2017*). These studies find that this micelle is remarkably stable, suggesting that completing fission faces a very sizable second free energy barrier. Additional tension might reduce its height, and indeed in vitro experiments by *Roux et al. (2006)* and *Morlot et al. (2012)* have found that longitudinal tension assists in producing fission (though it is interesting to note that our simulations present a tension-free pathway to fission). However, *Zhang and Müller (2017)* estimate that the barrier might remain as large as $30\,k_{\mathrm{B}}T$ even at the largest biologically justifiable tensions (close to uncontrolled rupture), which leads them to speculate that other factors help to complete fission, such as the high curvature in the region where the micelle merges with the bilayer.

Here we propose that this second barrier might indeed not be biologically relevant, since it is a consequence of the simulation setup in these studies, which prevents the micelle from shrinking and instead requires it to rupture mid-length. In our simulations we also never observe the micelle to break while being enclosed by the dynamin scaffold (from which it has already unbound), but we argue that this is due to the scaffold preventing the two point defects at the end of the micelle from merging. Once the scaffold disassembles (whether gradually or abruptly), they are being pulled together by a force that can be estimated to be $F \simeq \pi\kappa/2z_0 \sim \mathcal{O}(100\,\mathrm{pN})$. This is the energy per unit length of creating a cylindrical micelle of curvature radius $z_0$ (the pivotal plane distance) out of lipids that have a monolayer leaflet rigidity of $\kappa/2$. This force pulls the two daughter membranes together, whereupon bilayer contact catalyzes micellar fission (see *Video 6*). This hypothesis, even though derived from a fairly coarse-grained model, is nevertheless plausible, because splitting the cylindrical micelle somewhere along its length would further raise the free energy by creating two spherical caps of even more unfavorable packing geometry (*Zhang and Müller, 2017*), while the contact between the daughter membranes annihilates the two already existing point defects at the micellar ends.

The latter scenario seems to conflict with the widely held belief that highly curved vesicles are fusogenic. However, recent experiments by *François-Martin et al. (2017)* paint a more nuanced

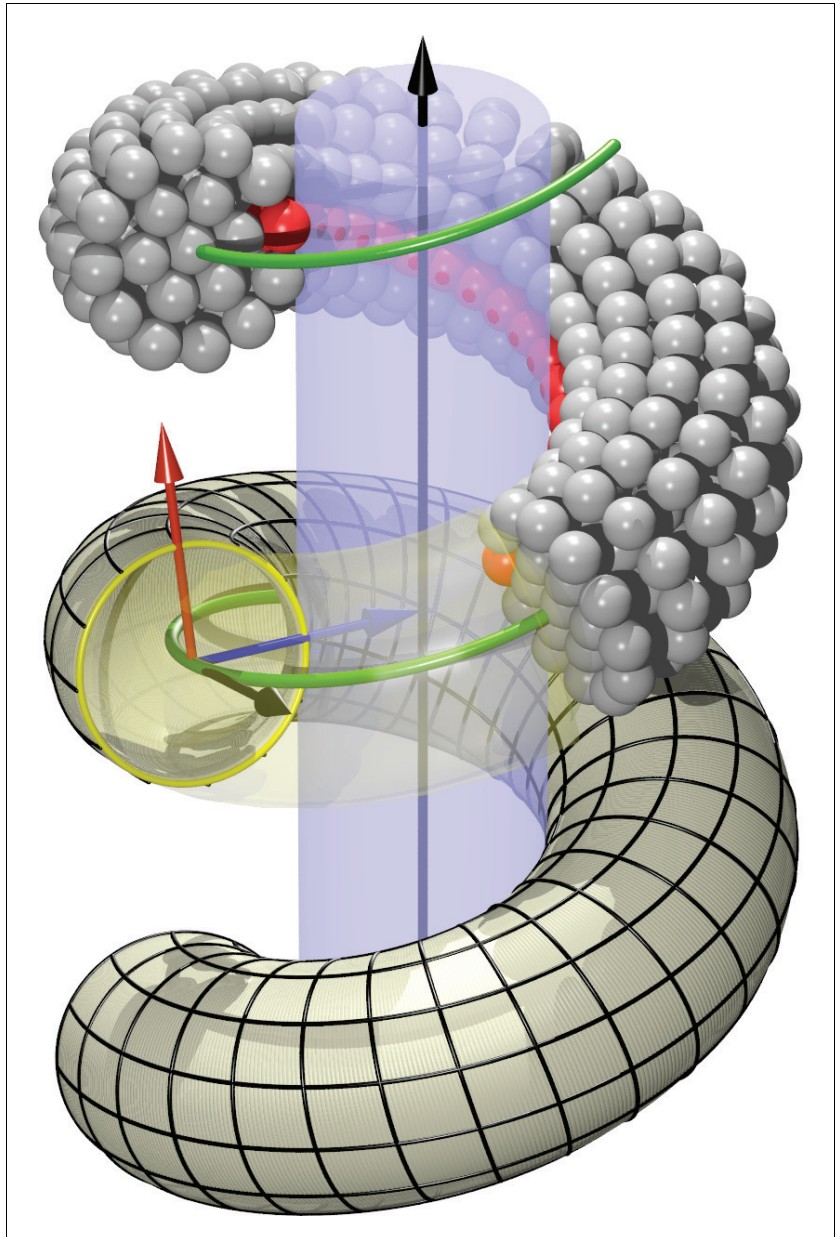

**Figure 5.** Definition of helicoidal coordinate system. The black vertical $z$-axis is surrounded by a cylinder, around which a helical dynamin tube winds. Its green center line with radius $R$ and pitch $2\pi p$ is given by (*Equation 1*). On this space curve we define the (right-handed) Darboux-frame $\{\boldsymbol{T}, \boldsymbol{N}, \boldsymbol{L}\}$, consisting of a tangent vector $\boldsymbol{T}$ (black), normal vector $\boldsymbol{N}$ (blue) and co-normal vector $\boldsymbol{L}$ (red), which are given by (*Equation 2*). In the $\boldsymbol{N}$-$\boldsymbol{L}$-plane we can then define the circular cross-section of the filament and place rings of CG beads at the correct distance from the green filament axis. The direction along $\boldsymbol{N}$ points towards the enclosed cylinder, but it is easy to rotate it by an angle $\varphi$ around the $\boldsymbol{T}$ axis. This is how one may rotate the beads representing the PH domain (red) off the underlying substrate.

DOI: https://doi.org/10.7554/eLife.39441.025

picture. These authors measured the free energy barrier towards fusion between $\sim 60\,\mathrm{nm}$ diameter POPC (16:0–18:1 PC) and DOPC [18:1 ($\Delta$9–Cis) PC] vesicles at 37°C. On the one hand, the observed free energy barrier of $\Delta F \sim 30\,k_\mathrm{B}T$ is at the lowest end of theoretical estimates, indicating that protein-free fusion between such vesicles is easier than previously thought. On the other hand, even when incubating such vesicles at the unusually high concentrations of 18 mM PC, only 2% had

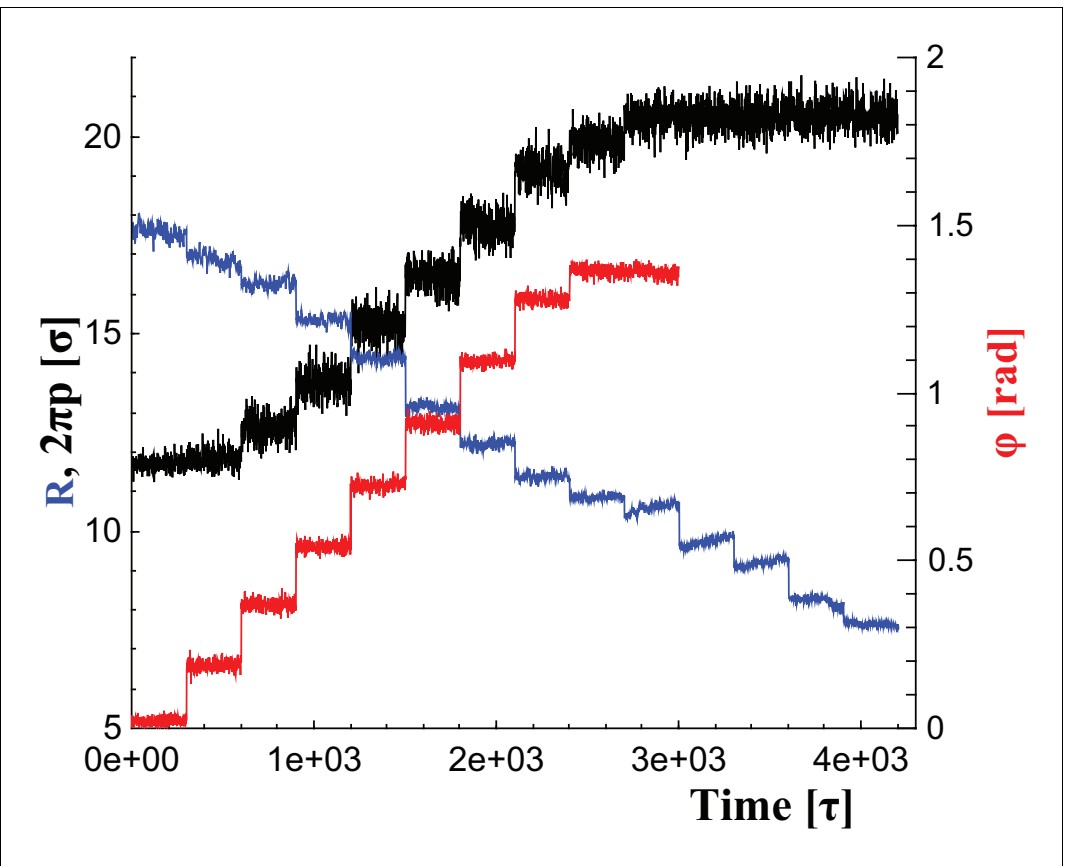

**Figure 6.** Changes of filament geometry. Time sequence followed when driving the dynamin filament through changes in its radius $R$, pitch $2\pi p$, and rotation angle $\varphi$. As detailed in the results section, we conduct sets of simulations in which various combinations of these observables are adjusted, while others stay at their original value. For instance, in a constriction+elongation simulation the radius is successively decreased from $18\sigma$ to $9\sigma$, while the pitch is simultaneously tuned up from $11.5\sigma$ to $20\sigma$ in a way documented by the blue and black curves, respectively, while the rotation angel remains at $\varphi = 0$.

DOI: https://doi.org/10.7554/eLife.39441.026

The following source data is available for figure 6:

**Source data 1.** Angle as a function of time.
DOI: https://doi.org/10.7554/eLife.39441.027
**Source data 2.** Pitch as a function of time.
DOI: https://doi.org/10.7554/eLife.39441.028
**Source data 3.** Radius as a function of time.
DOI: https://doi.org/10.7554/eLife.39441.029

completed fusion after 30 min, which does not support the notion of spontaneous fusion—as these authors indeed conclude.

Nevertheless, the free energy pathways for events that change membrane topology remain challenging, for at least four unrelated reasons: first, the choice of boundary conditions and constraints in theory or simulations can strongly affect the outcome (for instance, whether cylindrical micelles can shrink, hemifusion diaphragms can expand, or tensions can relax); second, the key physics happens at the nanometer scale, and hence the results of highly coarse-grained models such as ours or continuum theory must be interpreted cautiously; third, a topological barrier to fusion exists, whose height depends on the Gaussian curvature modulus and is thus not well known (see the contribution of Deserno in *Bassereau et al. (2018)*); and fourth, additional complexity due to accessory proteins or lipid mixtures may qualitatively change the energetics, for instance by lipid sorting in very high leaflet curvature gradients (*Cooke and Deserno, 2006*; *Tian and Baumgart, 2009*). For the pathway

that naturally arises in our simulations—a shrinking of the hemifission micelle post-disassembly, followed by autocatalytic cutting—the free energy barriers are not obvious, especially for solvent free models like ours (or those in the above-mentioned studies by *Mattila et al. (2015)* and *Zhang and Müller (2017)*), and hence this subject warrants more dedicated future studies.

## Results and discussion

Taken together, our studies support a number of conclusions. To begin with, the poppase model (*Stowell et al., 1999*) faces not only *kinetic* but also *equilibrium* obstacles: extension of the helical scaffold disfavors fission, because a widening groove offsets constriction. In contrast, hemifission is promoted by tilting one of the two symmetry subsets of PH domains (*Mehrotra et al., 2014*; *Sundborger et al., 2014*; *Kong et al., 2018*), which effectively rotates the filament's adhesive strip away from normal contact. The resulting Darboux torque (*Fierling et al., 2016*; *Guven et al., 2014*) has been previously invoked to explain fission (*Fuhrmans and Müller, 2015*; *Shnyrova et al., 2013*), but these earlier models pictured the dynamin scaffold as two rings, whose counter-rotation-induced torque remote-pinches the enclosed membrane tube. The effect is particularly intuitive under mirror symmetry, a geometry at odds with the actual helical one. Surprisingly, this difference appears not to be central.

We show that pre-hemifission scaffold breakage aborts fission by allowing the enclosed membrane tube to re-expand. But disassembly is still essential to complete fission, as argued in the two-stage model, because even if the severed bilayer is energetically preferable to the hemifission state (*Kozlovsky and Kozlov, 2002*), a substantial energy barrier is required to *explicitly* break the hemifission micelle (*Mattila et al., 2015*; *Zhang and Müller, 2017*). Our simulations lead us to suggest that this can be circumvented by merging and annihilating the micelle's two endpoint defects, a process that commences once the two daughter membranes are pulled into close proximity and hence critically depends on the scaffold getting out of the way by disassembling. Notice that this mechanism is not scaffold specific and may hence apply to any other fission machinery that leads to a hemifission state involving a small stretch of a cylindrical micelle.

How hemifission is induced is experimentally less clear (*Antonny et al., 2016*). In our studies constriction alone, down to the smallest experimentally observed luminal radii, does not suffice. We believe this to be independent of whether the constricted state is reached passively (two-stage model) or actively (constrictase model), since our dynamic protocol could simply be viewed as a means to adiabatically prepare a highly constricted state. PH domain tilting thus emerges as a way to catalyze hemifission, and since applying a Darboux torque costs energy, we posit that the energy of GTP hydrolysis is at least in part used to drive this conformational change. This is in accord with experimental findings by *Dar and Pucadyil (2017)* that replacing the PH domain by a simple binding motif strongly slows down the fission rate. It is also supported by the recent Cryo-EM reconstruction by *Kong et al. (2018)*, who suggest that forces generated from the GTPase dimer interaction are transferred across the stalk to the PH domain and from there onto the membrane. It is worth noting that in their reconstruction Dyn1 was bound to the nonhydrolyzable guanosine triphosphate analogue GMPPCP, and so the question whether GTP hydrolysis would create additional forces or torques that then could be transduced to the membrane remained open. To better understand the mechanical basis and viability of such a force transmission, it will be essential to structurally resolve the connection between the PH domain to the stalk domain.

Finally, we consistently observe that hemifission is initiated by a transient pore puncturing the constricted neck (see arrows in *Figure 2h*, the time sequence in *Figure 4*, and *Video 2*). This is reminiscent of likewise non-axisymmetric pathways seen in fusion simulations (*Noguchi and Takasu, 2001*; *Noguchi and Takasu, 2002*; *Müller et al., 2002*; *Müller et al., 2003*), whose implied transient leakage currents were soon after confirmed to be common in electrophysiological measurements on hemagglutinin-mediated fusion (*Frolov et al., 2003*). We estimate that our pores open to a diameter of no more than 2 nm, which under physiological ionic strength equips them with a conductivity on the order of 1 nS, while their lifetime is a few microseconds (mapped very roughly from our coarse-grained model via lipid self-diffusion). This would not be trivial to detect experimentally, but similarly fast transients (few microseconds) have been observed in the gating currents of Shaker potassium channels embedded in Xenopus oocytes, using an eight-pole Bessel filter and achieving a bandwidth of about 200 kHz (*Bezanilla, 2018*; *Sigg and Bezanilla, 2003*). Increasing the ionic strength in

reconstituted systems to 1 M, a further gain in sensitivity by up to an order of magnitude could be obtained, provided this does not interfere with dynamin's operation. But notice also that our leakage pores are partially covered by the dynamin scaffold (see again *Figure 2h*) and, under physiological conditions, possibly by additional proteins proposed to assist in dynamin-driven fission, such as BAR domains (*Takei et al., 1999*; *Farsad et al., 2001*; *Itoh et al., 2005*; *Mim et al., 2012*). Hence, a pore's effective conductance might be significantly lower than that of a freely accessible membrane pore or channel.

Our simulations show that pores are larger—and hence potential leakage more severe—if hemifission is triggered at larger radii, or if the filament is longer, suggesting that a close coordination of constriction, rotation, and possibly concomitant disassembly renders fission not only more *efficient* but also more *tight*. We hence expect mutants disrupting this coordination (for instance the ΔPH mutant of *Dar and Pucadyil (2017)*) to have larger pores. This would exacerbate leakage problems and could be experimentally observed.

While we have explicitly focused on the case of classical dynamin, several of our findings have implications beyond this particular protein and offer lessons for membrane topology remodeling that go beyond Dyn1. For instance, all members of the dynamin superfamily are believed to oligomerize (*Praefcke and McMahon, 2004*), likely into helices, for which our generic basic model holds. Furthermore, the mismatch between the normal curvature direction of such a helical filament and the direction of its adhesive domain (measured by $\varphi$ in our case) is a generic degree of freedom for such a structure. Whether directly present during assembly or only later triggered by a (possibly GTP-dependent) conformational change, it will give rise to Darboux torques on the membrane. We have explicitly shown that these support the transition into the hemifission state, and hence they provide a means to trigger a topological transition that is different from the notions of constriction or elongation. This matters because other topology remodeling proteins exist that are unlikely to work by either. Consider in particular the reverse-geometry scission driven by the ESCRT-III complex (*Wollert et al., 2009*), which involves the adsorption and polymerization of a helical filament on the *inside* of the neck to be cut. Present models (for a recent review, see *Schöneberg et al. (2017)*) rely on adhesion and/or geometric changes of the spiraling filaments, which result in either direct forces or boundary-induced stresses at the contact site (arising from, for instance, a contact curvature condition (*Deserno et al., 2007*)). But geometric rearrangements of a curved elastic filament with a finite twist rigidity, whose material frame is pinned to the membrane (*Guven et al., 2014*), almost invariably result in additional Darboux torques, as well as twist-induced analogs (*Fierling et al., 2016*), the relevance of which is only beginning to emerge (*Quint et al., 2016*). In this work we have explicitly demonstrated, for the particular example of Dyn1, that this geometrically elementary mechanism is remarkably effective, which suggest that it might be more common than so far realized: traces of the essential action could date back to the earliest bacterial FtsZ ancestor, which shares many of the key geometric (*Erickson, 2000*) and biochemical (*Lu et al., 2000*) features. This might hence suggest novel functional models also for other membrane remodelers (such as ESCRT-III), for which the mode of operation is much less well understood than for classical dynamin.

## Materials and methods

### General modeling aspects

Our investigation focuses on the mesoscopic effects of a constricting helical dynamin filament on a tubular lipid membrane, expressed primarily by the interplay between (*i*) filament geometry and binding affinity and (*ii*) membrane and filament elasticity. A long-standing aim in the dynamin field has been to develop an explanatory model for dynamin's membrane fission mechanism within the framework of such mesoscopic emergent properties. By creating a coarse-grained (CG) model of dynamin fission that captures precisely these mesoscopic aspects, our goal is to explore the consequences of this interplay, while either disregarding much of the microscopic or chemical detail, or implicitly accounting for it in terms of effective interactions.

The fundamental degrees of freedom in our model are mesoscopic 'beads' with a size (diameter) $\sigma \simeq 0.8\,\mathrm{nm}$, which hence correspond to $\mathcal{O}(10)$ heavy atoms. Physics below this scale leaves its trace at larger dimensions through the collective action of effective potentials—much like the quantum mechanics of correlated electron clouds re-emerges classically as effective van der Waals

interactions, or the configurational distributions of charged moieties in a molecule yield effective dipole moments and polarizabilities. A rich literature exists outlining the technique of coarse graining in a soft-matter and biophysics context (*Deserno, 2009*; *Ingólfsson et al., 2014*; *Izvekov and Voth, 2005a*; *Izvekov and Voth, 2005b*; *Brini et al., 2013*; *Müller-Plathe, 2002*; *Murtola et al., 2009*; *Noid, 2013*; *Noid et al., 2008*; *Peter and Kremer, 2009*; *Riniker et al., 2012*; *Saunders and Voth, 2013*; *Voth, 2008*). In the terminology of *Noid (2013)*, we employ a 'top down' modeling approach.

Since we are not concerned with hydrodynamic effects, we account for the embedding water implicitly via effective attractive interactions between hydrophobic species (such as CG lipid tail beads).

The reduction of the number of degrees of freedom allows not only for a computational speed-up and a better statistical sampling; it also offers insight into the nature of the original problem: physical effects that are correctly represented in a CG model prove to be largely independent of microscopic specifics, rendering them important players in an emergent mesoscopic explanatory model.

All simulations were performed using the ESPRESSO package (*Limbach et al., 2006*), using an integration time step $\Delta t = 0.01\,\tau$ (where $\tau$ is the simulations CG time unit). A Langevin thermostat (*Grest and Kremer, 1986*) with friction constant $\Gamma = 1.0\,\tau^{-1}$ was used to keep the temperature constant. All simulations have been performed in an $NVT$ ensemble.

## Lipid model

We use an implicit solvent lipid model (*Cooke et al., 2005*; *Cooke and Deserno, 2005*) in which a single lipid molecule is replaced by three consecutive beads, one for the hydrophilic head and two for the hydrophobic tails, which are not individually resolved (*Figure 1a*). In the absence of solvent, the hydrophobic effect, and hence aggregation of lipids into bilayer membranes, is driven by attractive interactions between the CG tail beads of depth $\varepsilon$ (the simulations' energy unit) and range $w_c$ (the precise potential forms are detailed by *Cooke et al. (2005)*). The magnitude of $w_c$ and the temperature $T$ determine whether lipids aggregate into fluid membranes, and if so, what their material properties are. We work at the frequently employed state point $w_c = 1.6\,\sigma$ and $k_B T = 1.1\,\varepsilon$ (which also sets the energy scale). Under these conditions lipids aggregate into fluid membranes with an area per lipid of about $a_\ell = 1.2\,\sigma^2 \approx 0.77\,\text{nm}^2$, an average head-bead distance from the midplane of about $d_H = 2.2\,\sigma \approx 1.8\,\text{nm}$, and a separation between the half-maximum density points (a possible proxy for the Luzzati thickness (*Luzzati and Husson, 1962*)) of about $d_{1/2} = 5.6\,\sigma \approx 4.5\,\text{nm}$. Considering the limitations of the underlying microscopic basis of nanometer-sized CG beads, these numbers are reasonably close to typical biologically relevant lipids, such as POPC ($a_\ell \approx 0.643\,\text{nm}^2$ and $d_{1/2} \approx 3.91\,\text{nm}$ at 30°C (*Kučerka et al., 2011*); still, sub-nanometer resolution should not be taken too literally in a model like this.

The membrane has a bending rigidity of $\kappa \approx 12.8\,k_B T$ (*Cooke et al., 2005*; *Cooke and Deserno, 2005*; *Harmandaris and Deserno, 2006*; *Hu et al., 2013*), purposefully chosen smaller than a typically value of $20\,k_B T$ (*Kučerka et al., 2005*), since this affords a significantly entropy-deprived coarse-grained system an alternative opportunity to undergo fluctuations. *Wang and Deserno (2016)*, recently also measured the tilt modulus of this model, finding it to be $\kappa_t \approx 7\,k_B T/\text{nm}^2$. This results in the characteristic tilt decay length $\ell_t = \sqrt{\kappa/\kappa_t} \approx 1.3\,\text{nm}$, in good agreement with experimental values (*Jablin et al., 2014*), indicating that the model does not merely capture overall fluidity and curvature elasticity, but also local lipid reorientation physics, which matters for the intermediate states of fission and fusion (*Kozlovsky and Kozlov, 2002*).

## Protein

We model the right-handed helical dynamin scaffold by CG beads of the same diameter $\sigma$ as the lipid CG beads (represented by a purely repulsive Weeks-Chandler-Andersen potential that only acts between filament and lipid beads), placing them such as to represent the shape of a CG filament (*Figure 1b*). This first requires setting up a coordinate system that embodies helical symmetry—see *Figure 5* for the following discussion.

The arc-length ($s$) parametrization of a right-handed helix of radius $R$ and pitch $2\pi p$, whose axis coincides with the $z$-axis of the coordinate system, is given by

$$X(s) = \begin{pmatrix} R\cos\left(s/\sqrt{R^2+p^2}\right) \\ R\sin\left(s/\sqrt{R^2+p^2}\right) \\ ps/\sqrt{R^2+p^2} \end{pmatrix}. \tag{1}$$

This has a length $2\pi\sqrt{R^2+p^2}$ per turn and a local (Frenet) curvature of $R/(R^2+p^2)$. The local unit *tangent vector* $T(s) = X'(s)$ is then

$$T(s) = \frac{1}{\sqrt{R^2+p^2}} \begin{pmatrix} -R\sin\left(s/\sqrt{R^2+p^2}\right) \\ R\cos\left(s/\sqrt{R^2+p^2}\right) \\ p \end{pmatrix}. \tag{2a}$$

We need to define a finite-thickness filament for which $X(s)$ is the central axis. Moreover, since we have to specify the location of an adhesive strip relative to a central cylinder that is being wrapped by the helix, it is convenient to extend the tangent vector $T$ along the helical curve into a local basis by defining two additional vectors $N$ and $L$ as follows: both have unit length and are perpendicular to $T$, the *normal vector* $N$ coincides with the local surface normal of the inscribed cylinder, and the remaining co-normal vector $L$ is given by $L = T \times N$:

$$\mathbf{N}(s) = \begin{pmatrix} -\cos(s/\sqrt{R^2+p^2}) \\ -\sin(s/\sqrt{R^2+p^2}) \\ 0 \end{pmatrix} \tag{2b}$$

$$\mathbf{L}(s) = \frac{1}{\sqrt{R^2+p^2}} \begin{pmatrix} p\sin\left(s/\sqrt{R^2+p^2}\right) \\ -p\cos\left(s/\sqrt{R^2+p^2}\right) \\ R \end{pmatrix}. \tag{2c}$$

The thus defined triplet $\{T, N, L\}$ of vectors constitutes a right-handed orthonormal basis at every point along the filament. It is called the 'Darboux frame' (with respect to the underlying cylindrical surface on which the filament rests).

On the $\{N, L\}$ plane perpendicular to the filament we now place CG beads that will form one of the cross-sectional circular discs from which we build the filament slice by slice. Each disk consists of a central bead placed directly onto the filament axis, a first ring of 6 beads and radius $R_6$, and a second ring of 12 beads and radius $R_{12}$. The coordinates of the beads sitting on the 1-, 6- and 12-ring can be parametrized as

$$\mathbf{X}_1(s) = \mathbf{X}(s) \tag{3a}$$

$$\mathbf{X}_6(s,n) = \mathbf{X}(s) + R_6[\mathbf{N}(s)\cos\varphi_n^{(6)} + \mathbf{L}(s)\sin\varphi_n^{(6)}] \tag{3b}$$

$$\mathbf{X}_{12}(s,n) = \mathbf{X}(s) + R_{12}[\mathbf{N}(s)\cos\varphi_n^{(12)} + \mathbf{L}(s)\sin\varphi_n^{(12)}] \tag{3c}$$

where $n \in \{0,\ldots,5\}$ for the 6-ring and $n \in \{0,\ldots,11\}$ for the 12-ring. The angles are given by

$$\varphi_n^{(6)} = \frac{2\pi n}{6} + \varphi \quad , \quad \varphi_n^{(12)} = \frac{2\pi n}{12} + \varphi , \tag{4}$$

and the overall phase shift $\varphi$ denotes the extent to which bead 0 on the 6- and 12-ring is rotated around the local filament axis, see *Figure 1b*. Since we will subsequently equip bead 0 on the 12-ring with an additional adhesion towards lipid head groups, representing the inner binding region of the dynamin filament due to the PH domains, this phase angle $\varphi$ describes a rotation of the CG filament with which we effectively capture the asymmetric tilting of half of the PH domains. Notice that due to the way the Darboux frame is set up, $\varphi = 0$ corresponds to an adhesive strip that exactly sits on the enclosed cylindrical surface (and hence does not exert any Darboux torque).

We typically consider filaments consisting of 9 dynamin dimers, each comprising five discs, for a total of 45 discs. We take $R_6 = 2\sigma$ and $R_{12} = 4\sigma$, making for a filament diameter of about $8\sigma \approx 6.4$ nm. The spacing $\Delta s$ between discs along the central helix is set so that a full turn in the unconstrained state $(R = 20\sigma$ and $2\pi p = 11\sigma)$ comprises 40 discs, leading to

$\Delta s = 2\pi\sqrt{R^2 + p^2}/40 = 3.1536\,\sigma$. The location of every bead in the filament at a given shape triplet $\{R, p, \varphi\}$ is now completely specified.

The shape is elastically fixed by introducing harmonic bonds between all beads within a cutoff distance of $r_{\mathrm{cut}} = 5\,\sigma$, whose rest length equals the equilibrium distance between the beads according to the chosen shape triplet $\{R, p, \varphi\}$, and whose spring constant is $K = 200\,\varepsilon/\sigma^2$, which is similar to the choice for other elastic networks for coarse-grained protein models (*Periole et al., 2009*). This renders the filament sufficiently stiff so that it can impose its shape on an underlying membrane tube, and not vice versa.

To change the filament's shape, we adjust the rest lengths of all elastic springs so that they represent distances in a filament at a new shape triplet, $\{R, p, \varphi\} \rightarrow \{R', p', \varphi'\}$. This introduces local stresses which the filament relaxes by transforming into the new equilibrium shape. We consider parameter ranges within $R/\sigma \in [6, 18]$, $2\pi p/\sigma \in [11, 20]$ and $\varphi \in [0°, 80°]$. *Figure 6* illustrates the time series of these parameter changes, which we use individually or in combination, as outlined in the main text. For instance, during constriction we reduce the helical radius by $1\,\sigma$ every $300\,\tau$, corresponding to about $0.18\,\mathrm{nm}/\mu\mathrm{s}$. Albeit much faster than in reality, this is still effectively quasistatic.

Beads on the red adhesive strip (accounting for the PH domains) additionally experience an adhesion of strength $\varepsilon$ towards lipid head beads, represented by a standard Lennard-Jones potential that is truncated and shifted to zero at $r = 2.5\,\sigma$. For the beads on the two immediately adjacent neighboring strips (i.e., numbers 1 and 11 on the 12-ring) we turn off the hard core repulsion with respect to lipid head (but not tail) beads, in order to permit the adhesive domain to embed into the head group region.

The free energy of binding for dynamin dimers or larger fragments depends not only on the local chemistry, but also on the curvature of both dynamin and the membrane. We are not aware of measurements that are precise enough to help parametrizing this interaction, but in order to avoid driving fission by overly strong interactions (which could, for instance, exert unrealistically large torques or even pull lipids out of the membrane), we have opted for a lower-bound scenario, in which we made the interaction between dimer-equivalents (blocks of 5 discs) and a highly curved membrane neck just strong enough to trigger binding; but when substrate curvature decreases due to membrane tube widening, the interplay between curvature energy and adhesion increasingly disfavors binding (*McDargh et al., 2016*), and fragments detach (*Video 3*).

## Measuring membrane constriction

Several possibilities exist to quantify the extent of tubular membrane constriction, but the two most obvious ones have significant drawbacks: the midplane radius cannot be defined once the membrane is in the hemifission state, and the radius of the inner lumen cannot distinguish between a completely closed bilayer tube and a hemifission micelle.

We hence use as a metric for constriction the cross-sectional *gyration radius* $R_{\mathrm{g}}$ of beads in the vicinity of the constriction point, defined as follows:

$$R_{\mathrm{g}}^2 = \frac{1}{M}\sum_{i=1}^{M}\left[(\boldsymbol{r}_i - \boldsymbol{r}_0)_{xy}\right]^2 , \tag{5}$$

where the sum extends over all lipids that are at most a radial distance of $20\,\sigma$ and an axial distance of $\pm 1\,\sigma$ away from the filament's center of mass, $\boldsymbol{r}_0$ is the center of mass of these selected lipids, and the subscript '$xy$' indicates that we first take the projection of $\boldsymbol{r}_i - \boldsymbol{r}_0$ into the $xy$-plane.

If $R_{\mathrm{m}}$ is the midplane radius of a cylindrical lipid tube and $w$ the monolayer width, then in continuum approximation we get

$$R_{\mathrm{g}}^2 = \frac{\displaystyle\int_{R_{\mathrm{m}}-w}^{R_{\mathrm{m}}+w} \mathrm{d}r\, 2\pi r\, r^2}{\displaystyle\int_{R_{\mathrm{m}}-w}^{R_{\mathrm{m}}+w} \mathrm{d}r\, 2\pi r} = R_{\mathrm{m}}^2 + w^2 , \tag{6}$$

showing that $R_{\mathrm{g}} = R_{\mathrm{m}}\left[1 + \frac{1}{2}(w/R_{\mathrm{m}})^2 + \cdots\right]$ is close to $R_{\mathrm{m}}$, with quadratic higher order corrections in $w/R_{\mathrm{m}}$. Moreover, a membrane cylinder of vanishing inner lumen has $R_{\mathrm{m}} = w$ and hence $R_{\mathrm{g}} = \sqrt{2}w$, while the hemifission micelle has $R_{\mathrm{g}} = w/\sqrt{2}$. Hence, the jump in $R_{\mathrm{g}}$ is $w/\sqrt{2} = \frac{1}{2} \times 5.6\,\sigma/\sqrt{2} \approx 2\,\sigma$, using

the Luzzati width of our CG membrane. This is very close to our observed jump distance in the constriction only case (*Figure 2f*, blue curve), which indeed only transitions when the inner lumen disappears. For the other cases we investigate the jump tends to be higher, because hemifission occurs while the membrane tube is still water-filled.

## Acknowledgments

We thank Marijn Ford, Aurelian Roux, Joshua Zimmerberg, Fred Lanni, Tina Lee, Adam Linstedt, Mathias Lösche, Martin Müller, and Antonio Raudino for discussions and critical comments. This work was supported by NSF Grants CHE #1464926, CHE #1764257, a Carnegie Mellon MCS/CIT Postdoctoral Fellowship, and the European Union's Horizon 2020 research and innovation program under the Marie Sklodowska-Curie grant agreement no. 754490.

## Additional information

### Funding

| Funder | Grant reference number | Author |
| --- | --- | --- |
| National Science Foundation | NSF CHE #1464926 | Markus Deserno |
| Carnegie Mellon University | Center of Excellence funding | Markus Deserno |
| Horizon 2020 - Research and Innovation Framework Programme | Marie Sklodowska-Curie grant agreement no. 754490 | Martina Pannuzzo |
| National Science Foundation | NSF CHE #1764257 | Markus Deserno |

The funders had no role in study design, data collection and interpretation, or the decision to submit the work for publication.

### Author contributions

Martina Pannuzzo, Conceptualization, Software, Formal analysis, Supervision, Funding acquisition, Investigation, Visualization, Methodology, Writing—original draft, Writing—review and editing; Zachary A McDargh, Conceptualization, Software, Formal analysis, Investigation, Visualization, Methodology, Writing—original draft; Markus Deserno, Conceptualization, Supervision, Funding acquisition, Investigation, Methodology, Writing—original draft, Writing—review and editing

### Author ORCIDs

Martina Pannuzzo (iD) http://orcid.org/0000-0001-8629-0173
Zachary A McDargh (iD) https://orcid.org/0000-0001-9022-5593
Markus Deserno (iD) https://orcid.org/0000-0001-5692-1595

### Decision letter and Author response

Decision letter https://doi.org/10.7554/eLife.39441.033
Author response https://doi.org/10.7554/eLife.39441.034

## Additional files

### Supplementary files

• Transparent reporting form
DOI: https://doi.org/10.7554/eLife.39441.030

### Data availability

The simulation software used is freely available at http://espressomd.org/wordpress/. Source data for Figure 2f, the supplement figure to Figure 2f, Figure 3, and Figure 6 are also provided.

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
