## [Decision Letter]

Thank you for submitting your article "The role of scaffold reshaping and disassembly in dynamin driven membrane fission" for consideration by *eLife*. Your article has been reviewed by Vivek Malhotra as the Senior Editor, a Reviewing Editor, and three reviewers. The following individuals involved in review of your submission have agreed to reveal their identity: Vadim Frolov (Reviewer #1); Aurélien Roux (Reviewer #2).

The reviewers have discussed the reviews with one another and the Reviewing Editor has drafted this decision to help you prepare a revised submission.

Summary:

The article describes the results of state-of-the-art CG simulations of membrane fission by dynamin, which, potentially, represent a considerable advance in understanding the mechanism of this phenomenon. The simulations allowed the authors to check, within the framework of their computational model, the feasibility of the previously suggested two-stage and constrictase scenarios of the process, to test whether the early suggested modes of the conformational changes of the dynamin helix can drive membrane fission, and to conclude that (i) the Darboux torque applied by the dynamin helix to the membrane is necessary for fission, (ii) the precursor of the hemi-fission stage is formation of a membrane nano-pore; (iii) the transition from the hemi-fission to the complete fission happens spontaneously after disintegration of the dynamin scaffold.

The reviewers found the results interesting and, potentially, suitable for publication in *eLife* provided that their comments are addressed.

Essential revisions:

1) One of the major concerns of the reviewers is a lack of the statistical validation of the results. In the current manuscript only one fission scenario is presented without any indication of the probability of its realization in the computational experiments.

Further, the statement that other scenarios, which include different combinations of the dynamin spiral constriction, elongation and rotation, are ineffective in terms of driving fission, needs a statistical substantiation.

A quantitative comparison of fission probability in all possible scenarios is necessary to validate the major conclusion of the study.

More specific questions would be: How variable is the fission pathway? How many hemi-fissions (in all cases? All with pores? Do pores ever go to complete fission directly?)? All of them are stable cylindrical micelles? All micelles shrink spontaneously to 0 length and then break upon the protein filament disassembly?

2) The quantitative characterization of the structural details of the suggested and alternative pathways has to be strengthened. Specifically,

- It is claimed that pores nucleating before hemi-fission are few nm wide and live for few microseconds, too small to be detected in real life. Of note, few nm pores are huge, not small (typical transient pores in electroporation are close/below 1 nm). Even if a pore opens for few microseconds, the integral charge transferred through it under a typical voltage bias of 100mV would be within detection limits. To substantiate the claim that the pores predicted by the proposed pathway are too small to be detected experimentally, determination of the size/open time distributions would be necessary.

In this context, the authors should comment on the validity of their strategy to simulate pores. In the case the computational method used is known to cause difficulties to simulate pore formation, it is requested that the authors perform simulations to estimate the pore formation at different tensions and compare the results to the experimental data (for example, Evans et al., 2003).

- It is claimed that pure constriction produces fission only upon complete closure of the lumen. What are the criteria for "no lumen"? How big is the difference (quantitatively, e.g. in nm) between "no lumen" and "visible" lumen seen in fissions caused by rotation+constriction? What are the variations in Rc? How Rc/its variations depend of the rotation angle/effective torque?

3) The agreement between the suggested mechanism and the existing structural data has to be further elaborated. Specifically:

- The Darboux torque is suggested to be produced when the dynamin adhesion line moves along a fixed/stable filament, e.g. via asymmetric displacement of the PH domains. This movement assumes synchronous tilting of many domains to produce the torque, yet such cooperative actions were seemingly ruled out. Then how does the torque build up? Further, can the PH domains support the torque providing that their connections to the protein stalk are rather flexible? More broadly, the torque creates stresses in the protein filament itself – is it realistic (can it be estimated) that the helical filament sustains the stress without changing shape?

Along the same lines, the authors state that tilting of the PH domain upon GTP hydrolysis could create a Darboux torque only if an asymmetric tilt occurs. In currently available cryo-EM maps (Sundborger et al., 2014), only 1 of the 2 PH domains of dimers is tilted, ensuring the required asymmetry. However, when looking at the tetrameric level, the titled PH domains are on each side of the tetramers, restoring symmetry. The authors should carefully check these points and explain how the data are compatible with their findings about the importance of the Darboux torque.

Finally, the tilting of the PH domains has been discussed in detail by biochemists, in particular, in the context of its effect on the molecular interactions between the membrane and the dynamin helix. Two levels of changes have been proposed: (i) tilting helps insertion of an amphipathic helix that helps to promote membrane curvature and thus constriction; (ii) titling breaks the specific lipid (PIP2)-dynamin bond or pulls the lipid out of the membrane. While the first change is unlikely because the amphipathic helix is located on the side of dynamin that is not in interaction with the membrane when titled (Harvey McMahon, private communication), the latter is essential for transmission of the Darboux torque to the membrane. It is thus required that the authors propose an estimate of the energy put on the dynamin/membrane bond when PH domains are tilting and compare it to the affinity values of dynamin PH domain for PIP2, and to forces required to pull off lipids from membrane.

4) The relationship with the results of other simulation models has to be thoroughly discussed. Specifically:

The conclusion that the scission of the stalk/micelle, may not be physiological relevant sounds problematic. Other simulation works show that a stalk formed between fusing vesicles is highly long living even for symmetric, membrane forming lipids such as POPC (Risselada, 2014). Those systems essentially do not differ from the ones simulated here after dynamin disassembly. In support of those results, it was recently found by one of the reviewers (using a string method) that a 'dimple' stalk formed between POPC membranes is thermodynamically stable and that scission would require a ~20 kBT barrier, quite in contrast to the here-envisaged fast rupture after Dynamin assembly. The question is whether the Crooks model underestimates the stalk stability and thereby falsely advocates the conclusion that the coined "second barrier" may not be physiological relevant. In flat membranes POPC stalks can become metastable and even stable under membrane dehydration (stabilization of the stalk is hydration repulsion driven in that case). The solvent free Crooks model cannot reveal such a behavior since it has a strong inherent tendency to form lamellar structures. Furthermore, the stalk may equally well expand after Dynanim disassembly, i.e., progression of fusion, since progression of fission and fusion are competitive pathways at this stage (they share the shame intermediate). The statement that small hemifused vesicles - which is essentially the structure one obtains after Dynamin disassembly - are poised to undergo fission is the exact opposite of the widely accepted observation that highly curved vesicles are fusogenic (even when being protein-free). In fact, there is evidence that completion of fission relies on feedback mechanisms and may involve several constriction cycles. This could suggest that scission attempts may fail, and that the mechanism perhaps relies on a dynamically imposed stress.

A related issue is the lack of movement of the (centers of mass of) daughter vesicles in Video 5 and Video 6 (comparable in size "protein pieces" move)? Is there a constraint? If so, does it affect the hemifission stability? Breakage of the cylindrical micelle upon shortening (Video 6) looks puzzling, one would rather expect formation of a stable stalk-like structure.

---

## [Author Response]

Summary:The article describes the results of state-of-the-art CG simulations of membrane fission by dynamin, which, potentially, represent a considerable advance in understanding the mechanism of this phenomenon. The simulations allowed the authors to check, within the framework of their computational model, the feasibility of the previously suggested two-stage and constrictase scenarios of the process, to test whether the early suggested modes of the conformational changes of the dynamin helix can drive membrane fission, and to conclude that (i) the Darboux torque applied by the dynamin helix to the membrane is necessary for fission, (ii) the precursor of the hemi-fission stage is formation of a membrane nano-pore; (iii) the transition from the hemi-fission to the complete fission happens spontaneously after disintegration of the dynamin scaffold.The reviewers found the results interesting and, potentially, suitable for publication in eLife provided that their comments are addressed.

We thank the reviewers for their positive assessment of our work and the many detailed and informed comments they have offered. We have made a number of significant changes to our manuscript, some of which required extensive additional simulations. Below we outline in detail how we have addressed the individual points that have been brought up.

Essential revisions:1) One of the major concerns of the reviewers is a lack of the statistical validation of the results. In the current manuscript only one fission scenario is presented without any indication of the probability of its realization in the computational experiments.Further, the statement that other scenarios, which include different combinations of the dynamin spiral constriction, elongation and rotation, are ineffective in terms of driving fission, needs a statistical substantiation.A quantitative comparison of fission probability in all possible scenarios is necessary to validate the major conclusion of the study.

We agree with the reviewer that additional statistical support of the simulation trajectories will indeed be beneficial. We have therefore repeated each of the four investigated studied fission scenarios (pure constriction, constriction+rotation, constriction+elongation,

constriction+rotation+elongation) in two additional simulation runs. The repeated runs confirm our overall conclusion but provide additional information on the variability of the process (see below). For the sake of visual clarity, we have not included all resulting 12 runs in a single figure but restricted to the original 4 representatives. However, we have included an additional supplementary figure which in each case show all executed runs and the resulting fission pathway.

Since in all cases we drive the system until it undergoes hemifission, the key question is not the “fission probability” but something more akin to an efficiency, namely, the question how far one has to constrict the neck to trigger the transition. In our work we find that the four scenarios we study rank in efficiency (from high to low) according to:

1) Constriction plus rotation

2) Constriction plus rotation plus elongation

3) Pure constriction

4) Constriction plus elongation

This ranking is consistent across all simulation runs, with cases (1) and (3) showing essentially no variation, and cases (2) and (4) some variation, which however does not switch the ranking. See also the discussion on variability below.

More specific questions would be: How variable is the fission pathway? How many hemi-fissions (in all cases? All with pores? Do pores ever go to complete fission directly?)? All of them are stable cylindrical micelles? All micelles shrink spontaneously to 0 length and then break upon the protein filament disassembly?

Let us begin with the reproducibility of our ranking: our repeated simulations show that the two cases of *pure constriction* and *constriction+rotation* show no statistically quantifiable variation. In each case hemifission happens at exactly the same amount of overall constriction (or constriction+rotation) as in the case we had previously documented. In contrast, the remaining two cases of *constriction+elongation* and *constriction+elongation+rotation* show some variability. In the first case, the additional two simulations transitioned even later into the hemifission state (meaning, requiring even more constriction and elongation than the case we had previously shown), while in the latter case one of the two simulations transitioned in the constriction step before and the other one in the step after the originally presented case.

All these findings are in accord with our previous claims: *constriction+elongation* is the least efficient pathway, and the two new simulations support this even more strongly. The other scenario, *constriction+elongation+rotation* is less efficient than *constriction+rotation*, but still more efficient than *pure constriction*, since the spectrum of variation still falls between these two bounds. We attribute the larger variability of these two pathways to the involvement of elongation, because this reduces the uniformity of overall confinement by opening the helical “grooves” along the dynamin scaffold. This enables larger conformational fluctuations of the enclosed lipid bilayer and hence leads to a larger variability. We now address all of this in our revised manuscript.

Next, the reviewers asked more specific questions on our pathways. Based on all available simulations we now have, based not just on three times as many constriction protocols as before, but also not forgetting that we have a variety of other protocols (such as disassembly, or stepwise disassembly), we can conclude the following:

1) We can only rigorously talk about pore formation in processes that have an inner lumen left, because otherwise it becomes somewhat ambiguous to talk about a pore (and, at any rate, it would not be experimentally measurable). That being said, that essentially means the case constriction+rotation, which in fact is the case that is physically realized. And in that case, we always see a pore, and that pore never directly proceed towards complete fission; we always first get a cylindrical micelle connecting the daughter vesicles.

2) None of our micelles ever break mid-length.

3) Micelles can only shrink spontaneously to zero length when the scaffold either has gotten out of the way, or concomitantly disassembles. But in these two cases, the micelle does spontaneously shrink. We would like to point out that anything else would be extremely surprising, since the energy per lipid in a micelle is significantly larger than the energy per lipid in a bilayer (this is why lipids make bilayers, not micelles), and so our observation that, given the choice, the lipids in a micelle physically in contact with a bilayer try to merge into the bilayer, is precisely what’s expected.

2) The quantitative characterization of the structural details of the suggested and alternative pathways has to be strengthened. Specifically,- It is claimed that pores nucleating before hemi-fission are few nm wide and live for few microseconds, too small to be detected in real life. Of note, few nm pores are huge, not small (typical transient pores in electroporation are close/below 1 nm). Even if a pore opens for few microseconds, the integral charge transferred through it under a typical voltage bias of 100mV would be within detection limits. To substantiate the claim that the pores predicted by the proposed pathway are too small to be detected experimentally, determination of the size/open time distributions would be necessary.

We believe this comment might have resulted from a misunderstanding, based likely on an unclear presentation of the relevant text in our manuscript. We expressly do not want to claim that these pores cannot be experimentally detected. Our intention was to give a rough idea what sizes and time scales we expect from these pores. Notice that given the nanometer scale of our observed pores, as well as the resolution of our coarse-grained model, it will be difficult in any case to pinpoint a precise length-scale. Moreover, the coarse-grained nature of the model also renders time-scale mapping problematic, and our estimated scale, based on lipid self diffusion as the scale-bar to translate between coarse-grained and real units, may be off (for instance, if a different dynamic process is more important). And finally, both the dynamin scaffold itself, as well as any other proteins, such as BAR domains, which will almost certainly be present in more realistic experimental situations, could “patch up” some of the leakage sites.

Given this situation, our primary concern was to suggest that the resulting pores are, in our estimation, small and short lived enough to render them not immediately visible in ordinary run-of-the-mill electrophysiology experiments. We would agree that more dedicated state-of-the-art experiments should be able to see these types of transient electrophysiological breakdowns. In fact, we would strongly hope that this is possible, since it would render our claim accessible to experimental tests.

We have significantly extended our discussion of the detectability of pores within the anticipated length- and time-scale window, suggesting that they should, indeed, be detectable, unless they are too well covered by the filament and other accessory proteins. Of course, our simple model cannot make any quantitative comment on the latter.

In this context, the authors should comment on the validity of their strategy to simulate pores. In the case the computational method used is known to cause difficulties to simulate pore formation, it is requested that the authors perform simulations to estimate the pore formation at different tensions and compare the results to the experimental data (for example, Evans et al., 2003),

We have in fact studied pore formation with the Cooke model in the past Cooke and Deserno, (2005). Specifically, we checked the pore opening scenario as a function of applied *strain*, which leads to an informative balance between membrane tension and the energy of an open pore (due to its rim). This particular setup is computationally more sensitive than controlling the applied tension, because the crucial transition at pore opening can be sampled without immediately thereafter destroying the membrane. One can make precise analytical predictions for the stress-strain relation of this pore opening scenario, which had previously already been discussed by both Farago, (2003) and Tolpekina, den Otter and Briels, (2004). This relation permits one to fit not only the area expansion modulus but also, crucially, the free energy per length of the open edge. Our results yield an edge tension of about 15pN and an (adiabatic!) rupture tension of about 8 mN/m, both of which are in very good agreement with typical values quoted for these observables. For full disclosure, this required fixing an analysis error in the original publication which led to an under-estimation of these two parameters by a factor of three - the correct numbers cited here were published in Deserno and Macromol. (2009). It is also worth noting that Tolpekina et al., had not just proposed the theory but also performed the same type of simulations we did in their work, using a slight better resolved (and explicit solvent) coarse-grained model due to Goetz and Lipowsky. Their results are qualitatively in agreement with ours, even though the value for their edge tension is about 2.7 times higher.

In the brief description of our lipid model at the end of the introduction we had listed a number of “local” properties which the Cooke membrane model reproduces well. We have included the pore opening scenario among that list and provided the references discussed above.

- It is claimed that pure constriction produces fission only upon complete closure of the lumen. What are the criteria for "no lumen"? How big is the difference (quantitatively, e.g. in nm) between "no lumen" and "visible" lumen seen in fissions caused by rotation+constriction? What are the variations in Rc? How Rc/its variations depend of the rotation angle/effective torque?

By “no lumen”, we mean a situation in which the distance *R*_i_ between the center of mass of the neck section (1nm wide along the *z*-direction, measured from the center of mass of the scaffold) and the top of the inner lipid head groups equals zero - see Author response image 1:

This distance is *very* challenging to quantify precisely, because it is very noisy, of the order of the CG bead size, and characterized by non-Gaussian fluctuations (due to the boundary nature of the lowest value “0”). Moreover, it does not permit an easy distinction between the state of zero lumen in an intact highly curved bilayer versus in a cylindrical micelle (in both cases there is no water channel left). This is why we instead use the radius of gyration, *R*_g_, as the cleaner observable to monitor the transition into hemifission.

For the case of rotation+constriction, the only one with a visible lumen, that visible lumen just before fissions measures about 1nm, and it is thus *not* vanishingly small. At this point a pore forms - by which we mean that the connection separating the inner and the outer lumen is broken. Figure 4 illustrates this transition. This sequence of images (with the time points relative to snapshot (a) as indicated) shows sections of the constricted neck, with the lipid headgroups rendered in blue and the lipid tails in yellow (the surrounding scaffold is not shown). The continuity of the tail region is emphasized in this image by using VMD’s “QuickSurf” rendering on all tail beads, which creates an isosurface extracted from a volumetric Gaussian density map; this visually emphasizes that all tails are part of a joint condensed aggregate, i.e., the bilayer region is intact. Snapshot (a) is taken moments before a pore opens, and a finite inner lumen is clearly visible. After the opening of a pore, visible in panel (b), the tail continuity is broken and a connection between the inner lumen and the space outside the neck opens up (indicated by the red arrow). Notice, however, that the inner lumen is still clearly distinct. In the subsequent snapshots (c) and (d) the pore widens circumferentially, but its upper and lower edge fuse with the inner leaflet of the inner lumen, thus again closing off the connection between outside volume and the interior of both daughter vesicles. The remaining snapshot (e) visualizes a cut through the cylindrical hemifission micelle. We have added this figure to our manuscript and discussed it in our text, in order to provide better visuals of the process.

In contrast, the transition into hemifission via constriction alone only occurs in our simulations when constriction eliminates the inner lumen (as defined above).

3) The agreement between the suggested mechanism and the existing structural data has to be further elaborated. Specifically:- The Darboux torque is suggested to be produced when the dynamin adhesion line moves along a fixed/stable filament, e.g. via asymmetric displacement of the PH domains. This movement assumes synchronous tilting of many domains to produce the torque, yet such cooperative actions were seemingly ruled out. Then how does the torque build up?

While it is true that many PH domains along the filament need to work together to create a large torque, synchronicity is not needed – it is certainly not a requirement for the geometric transformation we are proposing to be *feasible*. Our comments on a lack of cooperativity instead referred to the ATP hydrolysis, not the PH domain tilting. It is worth picturing the difference between the well-known putative mechanism in which neighboring rungs “step” along each other by some hydrolysis driven process, versus the tilting of PH domains on subsequent monomers. In the first case, cooperativity is crucial, because the filament can only move longitudinally if all pieces of the filament step together. Conversely, nothing requires the transverse tilting of PH domains to necessarily be synchronized. A maybe helpful image is this: if many people stand behind one another in a line, they can only move *forward* if they step together; but it is perfectly possible for every other person in the line to step *sideways* without requiring any such cooperativity.

How the torque is ultimately created is of course a crucial biochemical question, but one which we cannot answer in our coarse-grained model. It is interesting, though, that a recent paper from the Hinshaw lab (2018), which appeared after submission of our manuscript, explicitly describes the intrinsic post-assembly asymmetry in the dynamin dimer along the helical filament, which appears to be triggered by the interactions of G-domains (specifically following GTP binding, but prior to hydrolysis) and which is transmitted along one of the BSEs (by bending it) all the way down to the PH domain, triggering a substantial asymmetric force onto the membrane. We believe these findings further support our proposal of both asymmetry and the existence of Darboux torques, and we have added a discussion of this into our manuscript.

Further, can the PH domains support the torque providing that their connections to the protein stalk are rather flexible? More broadly, the torque creates stresses in the protein filament itself – is it realistic (can it be estimated) that the helical filament sustains the stress without changing shape?

It is obvious that the filament has to sustain some stress in order to fission the membrane. But then, this is no different in our proposal than in any previously proposed mechanism that involves constriction or elongation, or in particular the torque scenario advanced by Shnyrova, (2013), and for which the sustainability of a sufficient stress has, to our knowledge, been largely taken for granted - presumably out of necessity. Of course, this does not mean it is obvious, but it evidently can only be answered with a more atomistic underpinning of the dynamin filament as the basis. If we for instance were to envision an atomic-level elastic network model of the filament, this would almost by construction lead to the conclusion that a dynamin filament is a helical structure made from a material with a Young modulus approximately equal to that of hard plastic (as is true for virtually all compact globular proteins). Moreover, since for such materials the bending and the torsion moduli along cylindrical stretches are basically the same (up to minor Poisson ratio corrections), the filament would be able to withstand the kind of torsional stresses we describe, if it could withstand the bending stresses underlying the more traditional models.

If the reviewers are worried in particular about the rather flexible connection between the PH domain and the rest of the dynamin protein, then we would like to remark that flexibility does not necessarily exclude the ability to transmit torque - provided that other connections are also in place, which could simply be steric in nature. We reiterate that the recent cryo EM reconstruction from the Hinshaw lab (2018) provides very strong evidence that the asymmetric bending of the BSE transmits a force to the stalk, from there to the PH domain, and from there onto the membrane.

We now offer some of this discussion in the manuscript, to explain that the kind of stress resilience we require, while not obvious, is no more exotic than the corresponding mechanical assumptions underlying more traditional constriction or elongation models. Better numbers have to await dedicated atomistic modeling, which however is beyond the aim of our paper.

Along the same lines, the authors state that tilting of the PH domain upon GTP hydrolysis could create a Darboux torque only if an asymmetric tilt occurs. In currently available cryo-EM maps (Sundborger et al., 2014), only 1 of the 2 PH domains of dimers is tilted, ensuring the required asymmetry. However, when looking at the tetrameric level, the titled PH domains are on each side of the tetramers, restoring symmetry. The authors should carefully check these points and explain how the data are compatible with their findings about the importance of the Darboux torque.

When the reviewers refer to the “tetrameric level”, we assume they refer to the structure published by Reubold et al., (2015). We wish to emphasize that this structure has been determined in crystals without lipid, and it is not an assembled structure. If lipids are the cause (or at least contribute) to the asymmetry, that would not have been seen in this study. These authors then “copy-paste” that structure to the full helix, by matching the stalks. Again, if assembly should lead to something that breaks the symmetry, that would be lost in such a process. Hence, caution should be exercised when extrapolating from such crystal structures. Moreover, the full structural analysis by Sundborger on a helix assembled onto a lipid tube sees the persistent asymmetry, and so it is unclear why results derived from the lipid-less tetramer would be particularly pertinent. One might object that the analysis of their data might impose that each dimer is identical, and that would “lock in” the asymmetry. Still, this would nevertheless show that such a structure is compatible with the data. Moreover, the very recent paper from the Hinshaw lab already cited also sees the asymmetry very clearly, even though it is more obvious in the BSE. Hence, we believe that the existence of an asymmetry, on which the Darboux torque argument rests, is sufficiently well experimentally established.

Finally, the tilting of the PH domains has been discussed in detail by biochemists, in particular, in the context of its effect on the molecular interactions between the membrane and the dynamin helix. Two levels of changes have been proposed: (i) tilting helps insertion of an amphipathic helix that helps to promote membrane curvature and thus constriction; (ii) titling breaks the specific lipid (PIP2)-dynamin bond or pulls the lipid out of the membrane. While the first change is unlikely because the amphipathic helix is located on the side of dynamin that is not in interaction with the membrane when titled (Harvey McMahon, private communication), the latter is essential for transmission of the Darboux torque to the membrane. It is thus required that the authors propose an estimate of the energy put on the dynamin/membrane bond when PH domains are tilting and compare it to the affinity values of dynamin PH domain for PIP2, and to forces required to pull off lipids from membrane.

For the case of a neck between two vesicles (i.e., a locally approximately catenoidal geometry), we cannot easily produce an estimate of the energetic cost due to PH domain tilting, or the rotation of our helical scaffold. Such analyses have been performed for a much simpler system, namely a helical filament deforming a cylindrical membrane neck (McDargh and Deserno, 2018; Fierling et al., 2016); we wish to emphasize that these calculations made up the content of entire publications in their own right. Solving this problem for our twolobed membrane geometry would be much more difficult and is beyond the scope of this publication.

That being said, we wish to emphasize the following: the obvious worry in modeling this situation is that we overestimate the strength of scaffold binding, and therefore the extent to which the filament could transduce non-compressive forces (pull or shear) onto the underlying substrate. For this reason, when designing our model, we have decided to err - if anything – on the side of weak binding. More specifically, the attractive potential responsible for binding the scaffold to the membrane was chosen to be only strong enough so that short segments of the scaffold tend to bind to a pre-curved membrane rather than remaining in solution. This is essentially the weakest we can make the binding between the scaffold while ensuring that binding does occur. Notice that this is visible in the simulations in which we cut the filament into dimer-sized pieces but maintain the adhesion between these pieces and the lipid substrate (Video 3): as the neck region widens, or dimers diffuse into the less strongly curved lobes, these dimers have a tendency to unbind from the membrane. While this does of course not constitute a full free energy calculation, we believe it documents the most relevant point that our coarse-grained binding parameters have not been chosen unrealistically strong.

We had previously explained this choice in the protein modeling part of our Methods section, but we have now expanded the Discussion section to make the rationale more clear.

4) The relationship with the results of other simulation models has to be thoroughly discussed. Specifically:The conclusion that the scission of the stalk/micelle, may not be physiological relevant sounds problematic. Other simulation works show that a stalk formed between fusing vesicles is highly long living even for symmetric, membrane forming lipids such as POPC (Risselada, 2014). Those systems essentially do not differ from the ones simulated here after dynamin disassembly. In support of those results, it was recently found by one of the reviewers (using a string method) that a 'dimple' stalk formed between POPC membranes is thermodynamically stable and that scission would require a ~20 kBT barrier, quite in contrast to the here-envisaged fast rupture after Dynamin assembly. The question is whether the Crooks model underestimates the stalk stability and thereby falsely advocates the conclusion that the coined "second barrier" may not be physiological relevant. In flat membranes POPC stalks can become metastable and even stable under membrane dehydration (stabilization of the stalk is hydration repulsion driven in that case). The solvent free Crooks model cannot reveal such a behavior since it has a strong inherent tendency to form lamellar structures. Furthermore, the stalk may equally well expand after Dynanim disassembly, i.e., progression of fusion, since progression of fission and fusion are competitive pathways at this stage (they share the shame intermediate). The statement that small hemifused vesicles – which is essentially the structure one obtains after Dynamin disassembly – are poised to undergo fission is the exact opposite of the widely accepted observation that highly curved vesicles are fusogenic (even when being protein-free). In fact, there is evidence that completion of fission relies on feedback mechanisms and may involve several constriction cycles. This could suggest that scission attempts may fail, and that the mechanism perhaps relies on a dynamically imposed stress.

The reviewer points out an important issue that is indeed worth discussing more (as we now do in our manuscript), even though we will not be able to conclusively settle the question at this point. The challenge is that this is a highly complex problem that suffers from at least four independent difficulties:

1) Boundary conditions. The fate of a cylindrical lipid micelle connecting two vesicles, and the question whether the vesicles ultimately fuse or split, depends critically on the applied boundary conditions as well as other constraints imposed during a simulation. For instance, in the work by Zhang and Müller, (2017) the micelle is simply held at a given length and cannot shrink; something similar holds for the work by Risselada mentioned by the reviewer: we suspect that the boundary conditions applied in this work, which are different from ours, might prevent or at least delay the rearrangements needed to see the system progress to either fission or fusion. We don’t claim that our results contradict these studies; our point is that it is very difficult to compare them considering the important differences in constraints: when our vesicles separate, it’s neither a cylindrical micelle that’s breaking, nor a stalk. It’s a configuration in which a cylindrical micelle has essentially shrunk to zero length and now has the opportunity to annihilate two point defects against one another by splitting - a situation that is topologically different from the question whether a stalk (a single point defect) would split and complete towards fission. These two states may or may not be on different sides of an important free energy barrier, and in the absence of a detailed free energy study, which goes beyond the scope of our work, no premature conclusion about stability or the lack thereof can be drawn based on the outcomes of our simulations. Finally, notice that even in a case of free vesicles and no constraint on their separation, the situation would still be different from our simulation if one explicitly accounts for the solvent, because the associated constraint on volume translates into a constraint on tension via the Young-Laplace law, and then the tension relaxation will look quantitatively different.

2) Model resolution. Evidently, the question we are pondering here pertains to physics at the nanometer level, which is the scale at which we coarse grain. We have provided evidence that an important set of physical properties of the Cooke model work remarkably well down to that scale, and this is the reason why we even use it for the dynamin fission problem. However, there is evidently no guarantee that all properties, including even dynamical ones, come out right, and for that reason we would never rely on a model as coarse grained as this to proclaim that the local details of a fission or fusion process need to be rethought. All we do at this point is advance a hypothesis, and we now clearly mark it as such.

3) Topology. Bringing two vesicles together and fusing them creates an almost instant topological fusion barrier that depends on the value of the Gaussian curvature modulus, as one of us has recently pointed out (see the contribution by Deserno in the 2018 biomembrane curvature and remodeling roadmap Bassereau et al., (2018)). Since at present we know close to nothing about this modulus for any real system, it is extremely difficult to predict what kind of lipid will have what kind of topological contribution to a fusion process.

4) Idealizations. Real systems are of course even more complicated: even if we forget about the host of accessory proteins, we know that membranes consist of a mixture of lipids, and extremely high curvature regions can significantly sort them by, say, preferred curvature. This is, in fact, a phenomenon that has been observed and quantified with the Cooke model as one of its first applications: in Cooke and Deserno, (2006), Cooke and Deserno showed that while lipid spontaneous curvature is not enough to significantly sort lipids across curvature differences on the order of transport vesicle size (a claim subsequently experimentally confirmed by Tian and Baumgart, (2009)), they showed that the highly curved regions in budding necks can noticeably sort lipids.

Our comments that the second free energy barrier may not necessarily be relevant are largely independent of our specific simulation. We simply point out that the micelle need not break in the middle in order for fission to commence, since it also has two point defects to work with. In our discussions with many scientists since submission of this manuscript, we have not encountered anyone who would not consider that as a perfectly plausible alternative (Zhang and Müller in fact mention this in their paper). Our simulations indeed suggest that this is a possibility, even though we would never claim that we’ve “proved” that it works like that, considering the resolution of the Cooke model. We have now emphasized this caveat even more.

As far as the question goes whether the Cooke model is not able to show stable stalks because “it has a strong inherent tendency to form lamellar structures”, we are not sure where the reviewer takes the information from to make such a claim. We do not expect the model to be any more or less lamellar than other models, and the estimated spontaneous curvature of our lipids, the tilt fluctuations and their modulus, as well as the pivotal plane distance suggest that the model is no different in lamellarity than other models. Notice also that it correctly shows stable cylindrical micelles that refuse to break.

We thank the reviewer for reminding us of the common wisdom in the community that highly curved vesicles are fusogenic, even if protein-free. We have heard the same claims, but we think they are not as well established as their prevalence suggests. For instance, many protein-free spontaneous fusion assays use calcium as a means to get the membranes stick together (and help with dehydration). More importantly, in a very recent paper François-Martin, Rothman and Pincet, (2017) actually strive to quantify the free energy barrier towards protein-free fusion of small vesicles. They find about 30kT, which they say is at the lower end of estimates that have been proposed, and as such conclude that initiating fusion is easier than previously imagined. But looking at their experimental data, one can also see the following: in their assay, which contained thousands of highly curved vesicles at high concentration, about 2% of the vesicles had fused after a waiting time of 30 minutes. We would not want to describe this as highly fusogenic, and neither do these authors. Instead, they write:

“Finally, 30 kBT is an ideal value to enable facile membrane fusion as directed on demand in living cells: It will not happen spontaneously between bare membranes, yet as soon as specific fusion machinery is in place, it will be easily triggered.”

In other words, a fusion machinery is needed after all.

A related issue is the lack of movement of the (centers of mass of) daughter vesicles in Video 5 and Video 6 (comparable in size "protein pieces" move)? Is there a constraint? If so, does it affect the hemifission stability? Breakage of the cylindrical micelle upon shortening (Video 6) looks puzzling, one would rather expect formation of a stable stalk-like structure.

The very slow movement of the daughter vesicles is a direct consequence of the Langevin thermostat we employ to set the temperature. The presence of a friction force –ξ**v** on every bead in the vesicle (as well as independent Gaussian noise) means that even cooperative motions, in which all beads move together, are still penalized by friction. (The diffusivity reduction of the dynamin fragments is less pronounced than that of the daughter vesicles, because they contain a smaller number of CG beads.) This would be different in a DPD thermostat, in which only relative velocities enter. This, incidentally, is a vivid example how the dynamics of two simulation setups can differ even if they, by construction, represent exactly the same equilibrium thermodynamics, and it is the reason why we are very cautious when we refer to dynamical processes. That being said, the delay in the response of the daughters to the applied friction force is not physically unreasonable, because in the real situation there will be hydrodynamic friction, which is especially high if solvent needs to be expelled from narrow crevices. Even though our Langevin friction has a different origin, we hence feel that this particular way of thermalizing the system is still more natural than a DPD thermostat would be.

As far as the question is concerned whether in movie 6 the contracting cylindrical micelle should or should not form a stalk, we would like to refer the reviewer to our answer from the previous point. We do not think that the situation is as intuitively obvious as suggested. The cylindrical micelle state contains two point defect, which in a plane that contains the micelle look a bit like half defects in nematic liquid crystals. In contrast, a stalk is a single defect, which in the same plane would look like a -1 defect. (The analogy with topological defects is incomplete, but we do not wish to get sidetracked here.) It is absolutely not obvious that the merger of two -half defects into a -1 defect is downhill in elastic energy. In fact, in liquid crystals defects of the same sign repel. Here, they are drawn together by the tendency of the micelle to shrink an unfavorable energy per unit length, at the expense of drawing two singularities closer. What the free energy landscape for this looks like would be a highly fascinating project, for which we think the outcome is neither obvious nor uninteresting, but it would lead us far beyond the scope of the present manuscript to open up this question.